# Understanding the Limits of Unsupervised Domain Adaptation via Data Poisoning

**Akshay Mehra[1], Bhavya Kailkhura[2], Pin-Yu Chen[3] and Jihun Hamm[1]**
[1]Tulane University    [2]Lawrence Livermore National Laboratory    [3]IBM Research
{amehra, jhamm3}@tulane.edu, kailkhura1@llnl.gov, pin-yu.chen@ibm.com

## Abstract

Unsupervised domain adaptation (UDA) enables cross-domain learning without target domain labels by transferring knowledge from a labeled source domain whose distribution differs from that of the target. However, UDA is not always successful and several accounts of 'negative transfer' have been reported in the literature. In this work, we prove a simple lower bound on the target domain error that complements the existing upper bound. Our bound shows the insufficiency of minimizing source domain error and marginal distribution mismatch for a guaranteed reduction in the target domain error, due to the possible increase of induced labeling function mismatch. This insufficiency is further illustrated through simple distributions for which the same UDA approach succeeds, fails, and may succeed or fail with an equal chance. Motivated from this, we propose novel data poisoning attacks to fool UDA methods into learning representations that produce large target domain errors. We evaluate the effect of these attacks on popular UDA methods using benchmark datasets where they have been previously shown to be successful. Our results show that poisoning can significantly decrease the target domain accuracy, dropping it to almost 0% in some cases, with the addition of only 10% poisoned data in the source domain. The failure of these UDA methods demonstrates their limitations at guaranteeing cross-domain generalization consistent with our lower bound. Thus, evaluating UDA methods in adversarial settings such as data poisoning provides a better sense of their robustness to data distributions unfavorable for UDA.

## 1 Introduction

The problem of domain adaptation (DA) arises when the training and the test data distributions are different, violating the common assumption of supervised learning. In this paper, we focus on unsupervised DA (UDA), which is a special case of DA when no labeled information from the target domain is available. This setting is useful for applications where obtaining large-scale well-curated datasets is both time-consuming and costly. The seminal works [1, 2] proved an upper bound on a classifier's target domain error in the UDA setting leading to several algorithms for learning in this setting. Many of these algorithms rely on learning a domain invariant representation by minimizing the error on the source domain and a divergence measure between the marginal feature distributions of the source and target domains. Popular divergence measures include total variation distance, Jensen-Shannon divergence [9, 32, 37], Wasserstein distance [29, 15, 7], and maximum mean discrepancy [19, 17, 20]. The success of these algorithms is argued in terms of minimization of the upper bound proposed in [1] along with an improvement in the target domain accuracy on benchmark UDA tasks.

Despite this success on benchmark datasets, some works [9, 16, 36, 33] have presented evidence of the failure of these methods in different scenarios. Recent works have explained this apparent failure of UDA methods by proposing new upper bounds [36, 6, 14, 34] on the target domain error while

35th Conference on Neural Information Processing Systems (NeurIPS 2021).

others have demonstrated the cause of failure using experiments showing that learning a domain invariant representation that minimizes the source domain error can cause an increase in the error of the ideal joint hypothesis [16]. To provably explain this apparent failure of learning in the UDA setting we propose a lower bound on the target domain error. Our lower bound provides a necessary condition for successful learning in the UDA setting, complementing the existing upper bound of [1] and is dependent on the difference between the labeling functions of source and target domain data induced by the representation map. For cases where the induced labeling functions match on the source and the target domain data (i.e., favorable case), the success of UDA is explained using the upper bounds proposed by previous works [1, 36, 14]. For a representation that aligns the source and the target domain data and minimizes the error on the source but induces labeling functions that don't agree on the source and the target domain data (i.e., unfavorable case), our lower bound explains the failure of UDA. Our analysis brings to light yet another case (i.e., ambiguous case) of data distributions where success and failure of UDA are equally likely. This happens due to the lack of label information from the target domain. This case opens doors for adversarial attacks against UDA methods since a small amount of misinformation about the target domain labels can lead the UDA methods into producing a representation similar to the unfavorable case, incurring a significant increase in the target domain error.

Motivated from this analysis of UDA methods under different data distributions, we evaluate the extent to which the performance of current UDA methods can suffer in presence of a small amount of adversarially crafted data. For this purpose, we propose novel data poisoning attacks, using mislabeled and clean-label points. We evaluate the effect of our poisoning attacks on popular UDA methods using benchmark datasets, where they were previously shown to be very effective. We find that our poisoning attacks cause UDA methods to either align incorrect classes from the two domains or prevent correct classes from being very close in the representation space. Both of these lead to the failure of UDA methods at reducing target domain error. With just 10% poison data in the source domain, target domain accuracy for current UDA methods is significantly reduced, dropping to almost 0% in some cases. This dramatic failure of UDA methods demonstrates their limits and suggests that the future UDA methods must be evaluated in adversarial settings along with evaluation on benchmark datasets to truly gauge their effectiveness at learning under the UDA setting.

Our main contributions are summarized as follows:

- We prove a lower bound on the target domain error that provides a necessary condition for successful learning in the UDA setting. Our bound shows the failure of learning a domain invariant representation while minimizing the source domain error at guaranteeing target generalization.

- We present example data distributions where UDA succeeds, fails, and where success and failure are equally likely. This sensitivity of UDA methods to the data distribution brings to light a new vulnerability of UDA methods to adversarial attacks such as data poisoning.

- To concretely understand the extent of this vulnerability of UDA methods, we propose novel data poisoning attacks using clean-label and mislabeled data. Our results show a dramatic failure of current UDA methods at target generalization in presence of poisoned data. Thus, our poisoning attacks can provide better insights into the robustness of UDA methods than those obtained from performance evaluation on benchmark datasets.

## 2 Background and related work

**Analysis of unsupervised domain adaptation:** Several previous works have studied the problem of UDA and have provided conditions under which UDA is possible [2, 1, 21, 22, 8, 3, 34]. [1, 2] proposed an upper bound on the target domain error which has inspired many UDA algorithms. Recent works [36, 6, 14] have improved the upper bounds on target domain error and have proposed a lower bound dependent on the labeling functions in the input space. Using this lower bound, the failure of learning in the UDA setting was explained in the case when marginal label distributions are different between the two domains. Our lower bound, on the other hand, is dependent on the labeling functions for the source and target domains, induced by the representation. This suggests that if a representation induces labeling functions that disagree on source and target domains then UDA provably fails even if the representation is domain invariant and minimizes error on the source domain. Thus our lower bound directly explains the observations of failure of UDA in many previous

works [9, 16, 33, 36, 34]. A detailed comparison of our work with other works analyzing the failure of learning in the UDA setting is present in Appendix G.

**Algorithms for UDA:** Many algorithms [9, 32, 18, 37, 30, 10] for UDA learn a domain invariant representation while minimizing error on the source domain. A popular adversarial approach to domain adaptation is DANN [9] which uses a discriminator to distinguish points from source and target domains based on their representations. Another popular method CDAN[18], uses classifier output along with representations to identify the domains of the points. IW-DAN and IW-CDAN [6] were recently proposed as extensions of the original DANN and CDAN with an importance weighting scheme to minimize the mismatch between the labeling distributions of the two domains. A different approach MCD [27], makes use of two task-specific classifiers as discriminators to align the two domains. This method adversarially trains the representation to minimize the disagreement between the two classifiers on the target domain data (classifier discrepancy) while training the classifiers to maximize this discrepancy. Another recent approach, SSL [35] uses self-supervised learning tasks (e.g. rotation angle prediction) to better align the two domains. In this work, we study the effect of poisoning on these methods as they have been shown to be effective at various UDA tasks.

**Data poisoning:** Data poisoning [4, 25, 12, 5, 13, 31, 24] is a training time attack where the attacker has access to the data which will be used by the victim for training. Most works [38, 26, 28, 23, 11] have considered data poisoning in a fully supervised setting where train and test sets are drawn from the same underlying data distribution (single domain setting). These works either target the classification of a single test point by adding a large number of poisoned data or require modifying all points in a class to affect the model's performance on that class after retraining. Our work studies the effect of poisoning on the entire target domain in the UDA setting using popular UDA algorithms for training. The success of our poisoning attacks at significantly reducing the target domain accuracy with a small amount of poisoned data shows the ease of poisoning in the UDA setting. In contrast, poisoning leads to a small decrease in the overall test accuracy in the single domain setting, especially when using state-of-the-art classifiers such as deep neural networks.

# 3 When does learning fail in the unsupervised domain adaption setting?

**Notations and settings:** Let $\mathcal{X}$ be the data domain and $\mathcal{D}$ be the distribution over $\mathcal{X}$ with the corresponding pdf $p(x)$. We assume there is a deterministic labeling function $f : \mathcal{X} \to [0, 1]$ for the given binary classification task. The $f(x)$ can be interpreted as $Pr[y = 1|x]$. Let $g : \mathcal{X} \to \mathcal{Z}$ denote the representation map that maps an input instance $x$ to its features where $\mathcal{Z}$ is called feature or representation space. Let $h : \mathcal{Z} \to [0, 1]$ be a hypothesis for binary classification on the representation space. Note that the representation map $g$ induces a distribution over $\mathcal{Z}$ denoted by $Pr_{\tilde{\mathcal{D}}}[B] := Pr_{\mathcal{D}}[g^{-1}(B)]$ and the corresponding density function $\tilde{p}(z)$ on $\mathcal{Z}$ [2]. The $g$ also induces the labeling function

$$\tilde{f}(z) := E_{\mathcal{D}}[f(x)|g(x) = z], \tag{1}$$

for any $B$ such that $g^{-1}(B)$ is $\mathcal{D}$-measurable. The misclassification error $e(h)$ w.r.t. the induced labeling function is $e(h) = E_{z \sim \tilde{\mathcal{D}}}[|\tilde{f}(z) - h(z)|]$ where $\tilde{\mathcal{D}}$ is the induced distribution over $\mathcal{Z}$. Similarly, we define $e(\tilde{f}, \tilde{f}') = E_{z \sim \tilde{\mathcal{D}}}[|\tilde{f}(z) - \tilde{f}'(z)|]$ and $e(h, h') = E_{z \sim \tilde{\mathcal{D}}}[|h(z) - h'(z)|]$. The distributions $\tilde{p}$ and the labeling functions $\tilde{f}$ for the source and the target domains will be written as $\tilde{p}_S, \tilde{p}_T, \tilde{f}_S$ and $\tilde{f}_T$, respectively. The total variation distance is $D_1(\tilde{p}, \tilde{p}') = \int_{\mathcal{Z}} |\tilde{p}(z) - \tilde{p}'(z)| dz$.

## 3.1 Lower bound on the target domain error

Most adversarial DA methods learn a domain invariant representation $g : \mathcal{X} \to \mathcal{Z}$ by minimizing error on the source domain and penalizing the mismatch between the **marginal** source and target distributions since the conditional distribution $\tilde{p}_T(z|y)$ for target domain is unavailable in the UDA setting. Some works [33, 36, 16] have shown this to be insufficient at guaranteeing target generalization. Recent works have proposed a new upper bound [36] or argued failure in terms of the upper bound in [1, 2] ($e_T(h) \leq \min\{e_T(\tilde{f}_S, \tilde{f}_T), e_S(\tilde{f}_S, \tilde{f}_T)\} + e_S(h) + D_1(\tilde{p}_S, \tilde{p}_T)$) being large. However, a large upper bound does not guarantee failure. Thus, we prove a simple lower bound on the target domain error which shows the necessary condition for the success of learning in the UDA setting and also explains the failure of current UDA methods at guaranteeing target generalization.

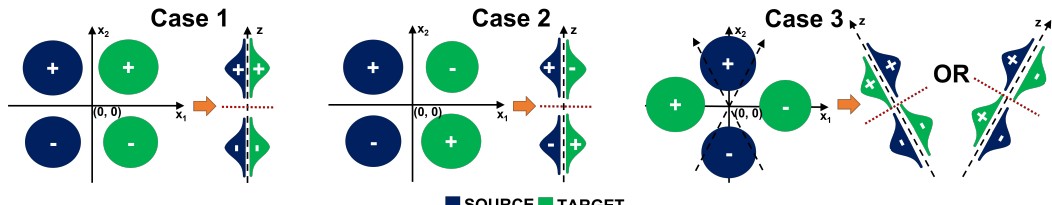

Figure 1: Illustrative cases for UDA. In UDA, one can observe the source data (blue blobs), the source labels (class + and -), and the target data (green blobs) but not the target labels. The optimal decision boundary (red dotted line) and linear representation $g(x) = u^T x$ from the input space in $\mathbb{R}^2$ to the feature space in $\mathbb{R}$ (dashed line) minimizing the source and the alignment losses in Eq. 3 can be computed accurately. Depending on the data distribution, domain adaptation can be successful (Case 1), fail (Case 2), or be undetermined (Case 3). Case 3 has two global minima with drastically different target domain performance, even though representations in both cases minimize the error on the source distribution as well as align the marginal distributions of the two domains.

**Theorem 1.** *Let $\mathcal{H}$ be the hypothesis class and $\mathcal{G}$ be the class representation maps. Then, for all $h \in \mathcal{H}$ and $g \in \mathcal{G}$,*

$$e_T(h) \geq \max\{e_S(\tilde{f}_S, \tilde{f}_T), e_T(\tilde{f}_S, \tilde{f}_T)\} - e_S(h) - D_1(\tilde{p}_S, \tilde{p}_T). \tag{2}$$

The proof is in Appendix A. Even though our bound depends on the total variation distance which is a strict measure of distance and is difficult to estimate, the bound still explains the limitations of UDA. Extending the bound to different divergence metrics is left as future work. The following corollary is immediate from Theorem 1.

**Corollary 1.1.** *For all $h \in \mathcal{H}$, $g \in \mathcal{G}$,*

$$e_T(h) \geq \max\{e_S(\tilde{f}_S, \tilde{f}_T), e_T(\tilde{f}_S, \tilde{f}_T)\} - e_S(h) - \sqrt{0.5 D_{KL}(\tilde{p}_S || \tilde{p}_T)}.$$

This is due to the Pinsker's inequality: $D_1(p, p') \leq \sqrt{0.5 D_{KL}(p||p')}$. The interpretation of Eq. 2 is as follows. Since $f_T$ is not observable, the goal of UDA is to minimize the observable source classification error $e_S(h)$ and the domain mismatch $D_1(\tilde{p}_S, \tilde{p}_T)$ (or other metrics such as $d_{\mathcal{H}\Delta\mathcal{H}}(\tilde{p}_s, \tilde{p}_T)$):

$$\min_{g,h} \ e_s(h) + D_1(\tilde{p}_S, \tilde{p}_T). \tag{3}$$

This leads to maximization of the second and the third term in the RHS of Eq. 2. With overparameterized models and large datasets, minimizing the empirical estimates of the quantities in Eq. 3 can drive $e_S(h) \simeq 0$ and $D_1(\tilde{p}_S, \tilde{p}_T) \simeq 0$. Consequently, $e_T(h) \geq \max\{e_S(\tilde{f}_S, \tilde{f}_T), e_T(\tilde{f}_S, \tilde{f}_T)\}$. If $\tilde{f}_S$ and $\tilde{f}_T$ disagree on the source and target domains in the representation space, target domain error $e_T(h)$ will be **provably** large.

**Corollary 1.2.** *For all $h \in \mathcal{H}$ and $g \in \mathcal{G}$,*

$$|e_T(h) - e_S(\tilde{f}_S, \tilde{f}_T)| \leq e_S(h) + D_1(\tilde{p}_S, \tilde{p}_T), \ \text{ and } \ |e_T(h) - e_T(\tilde{f}_S, \tilde{f}_T)| \leq e_S(h) + D_1(\tilde{p}_S, \tilde{p}_T).$$

This is obtained by combining the upper and the lower bounds (Appendix A). Thus a UDA method can only guarantee the target error $e_T(h)$ to be close to the labeling function mismatch $e(\tilde{f}_S, \tilde{f}_T)$. Whether $e(\tilde{f}_S, \tilde{f}_T)$ becomes larger or smaller after solving Eq. 3 is data/model dependent. For concreteness, we analyze the sensitivity of the performance of UDA methods to different data distributions using the following illustrative examples.

### 3.2 Illustrative examples showing the sensitivity of UDA methods to data distributions

Assume mixture-of-Gaussian distributions $p_S(x)$ and $p_T(x)$ for the source and the target domains in the input space, shown as blue and green blobs in Fig. 1. We consider a linear representation map $g(x) = u^T x$ from the input space $\mathcal{X} \subset \mathbb{R}^2$ to the feature space $\mathcal{Z} \subset \mathbb{R}$ (dashed line) that minimizes the source classification loss plus the marginal mismatch loss in Eq. 3. The optimal solution $g$ to the minimization problem can be found accurately using a mix of analytical and

numerical optimizations (detailed in Appendix B). We demonstrate the performance of UDA on three different data distributions. In Case 1 (favorable case), the true labeling function for source and target is $f_S((x_1, x_2)) = f_T((x_1, x_2)) = I[x_2 \geq 0]$, that is, $f$ is 1 in the upper halfspace and 0 in the bottom halfspace. One can verify (Appendix B) that the optimal $u$ is the vertical direction ($u = [0, 1]^T$) and the best hypothesis is $h(z) = I[z \geq 0]$, in which case $e_S(h) = 0$ and $D_1(\tilde{p}_S, \tilde{p}_T) = 0$. That is, perfect source classification and a perfect alignment of the marginals are achieved. Furthermore, the true labeling function in $\mathcal{Z}$ is $\tilde{f}_S(z) = \tilde{f}_T(z) = I[z \geq 0]$ (from Eq. 1) and therefore $e(\tilde{f}_S, \tilde{f}_T) = 0$ as well as the target loss $e_T(h) = 0$. In other words, the representation that minimizes Eq. 3 simultaneously **minimizes** $e(\tilde{f}_S, \tilde{f}_T)$, achieving the goal of reducing $e_T(h)$. However in Case 2 (unfavorable case), the true labeling function for target is upside-down $f_T((x_1, x_2)) = I[x_2 \leq 0]$. Since UDA does not use the target label nor the true labeling function, the optimal $g$ is exactly the same as Case 1 ($u = [0, 1]^T$), in which case we still have $e_S(h) = 0$ and $D_1(\tilde{p}_S, \tilde{p}_T) = 0$, but the labeling function mismatch becomes $e(\tilde{f}_S, \tilde{f}_T) = 1$ as well as $e_T(h) = 1$ which is the worst case. In other words, the representation that minimizes Eq. 3 simultaneously **maximizes** $e(\tilde{f}_S, \tilde{f}_T)$, totally failing at the goal of reducing $e_T(h)$. Case 3 exemplifies an ambiguous case. The optimal projection minimizing $e_S$ is still the vertical direction $u = [0, 1]^T$ but the optimal projection minimizing the alignment loss $D_1(\tilde{p}_S, \tilde{p}_T)$ can be either of the $\pm 45°$ directions ($u = [\pm 1/\sqrt{2}, 1/\sqrt{2}]^T$) with no preference of one over the other. Therefore the optimal solution $u$ for Eq. 3 that trades off the source error and the mismatch loss has two equal-valued global solutions $[\pm u_1, u_2]^T$ for some $u_1, u_2$. One solution (Fig. 1, Case 3, left) yields a small $e(\tilde{f}_S, \tilde{f}_T)$ and the other (Fig. 1, Case 3, right) yields a large $e(\tilde{f}_S, \tilde{f}_T)$. As can be intuitively seen, UDA is successful in the former but fails in the latter, and which equal-valued solution will be chosen is undetermined. A similar example of the failure of UDA methods due to the presence of two global optima was presented in [14]. Unlike their example, we use mixture-of-Gaussian distributions and analytically compute the representation that will be obtained from the minimization of Eq. 3. Using the obtained representations, we evaluate the quantities appearing in our lower bound and show that it is predictive of the performance of UDA methods on different data distributions. Details of the analysis and the results are in Appendix B.

Motivated from this extreme sensitivity of the performance of UDA methods on simple data distributions, we set out to explore the effects of small changes in real-world data distributions (via data poisoning) on state-of-the-art UDA methods. The results of our poisoning attacks in the next section suggest that slight changes in the data distributions could have a drastic impact on the target domain generalization performance of state-of-the-art UDA methods. Our findings highlight the importance of using adversarial settings (such as evaluating the performance of UDA methods in presence of poisoned data) to gauge the effectiveness of future UDA methods at learning in the UDA setting.

## 4 Breaking unsupervised domain adaptation methods with data poisoning

In this section, we present our novel data poisoning attacks to evaluate the ease with which current UDA methods can be fooled into producing a representation that leads to a large error on the target domain[1]. We propose three methods to generate poisoned data which will be added to the clean source domain data. The first poisoning attack uses mislabeled data as poisons. Under this attack, we evaluate two approaches (a) adding mislabeled source domain data (wrong-label correct-domain poisoning) and (b) adding mislabeled target domain data (wrong-label incorrect-domain poisoning) as poisons. The second attack adds images from the source domain watermarked with images from the target domain with incorrect labels (watermarking attack) as poisons. The last poisoning attack uses poisoned data with clean labels. We evaluate two approaches for this attack (a) using source domain data (clean-label correct-domain poisoning) and (b) using target domain data (clean-label wrong-domain poisoning) to initialize the poison data. The intuitive pictures of how these poisoning attacks hurt UDA methods are shown in Figs. 4, 5, and 6 (details in Appendix C). We evaluate the effect of poisoning on popular UDA methods namely, DANN[9], CDAN[18], MCD[27], SSL[35] (with rotation-angle prediction task), IW-DAN[6], and IW-CDAN[6]. We compare the difference in the target accuracy attainable by these methods when using clean versus poisoned data. Two benchmark datasets are used in our experiments, namely Digits and Office-31. We evaluate four tasks using SVHN, MNIST, MNIST_M, and USPS datasets under Digits and six tasks under the Office-31 using Amazon (A), DSLR (D), and Webcam (W) datasets. Additional experiments evaluating the

---

[1]Our code can be found at `https://github.com/akshaymehra24/LimitsOfUDA`.

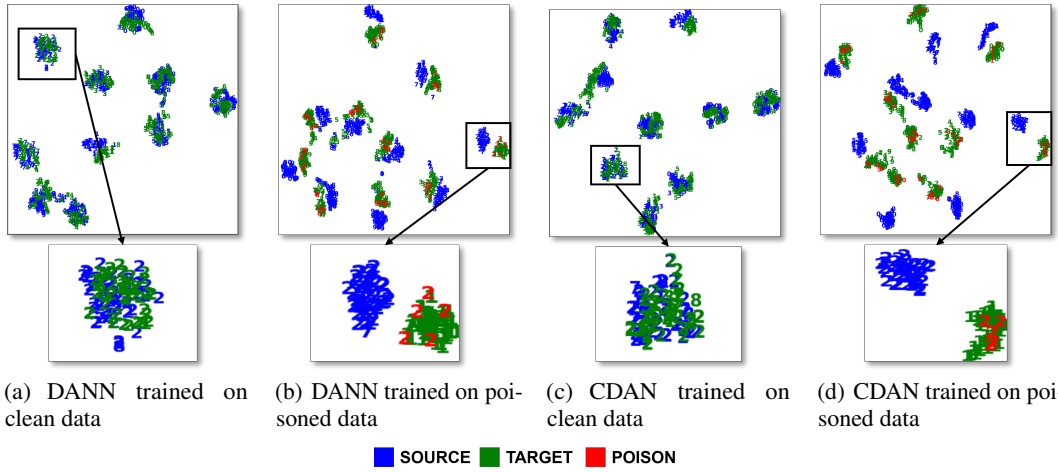

(a) DANN trained on clean data

(b) DANN trained on poisoned data

(c) CDAN trained on clean data

(d) CDAN trained on poisoned data

■ SOURCE ■ TARGET ■ POISON

Figure 2: (Best viewed in color.) t-SNE embedding of the data in the representation space (for MNIST → USPS task) learned using DANN and CDAN on clean and poisoned source domain data. Without poisoning, correct classes (data from source class 2 is zoomed in) from two domains are aligned ((a) and (c)). The presence of poisoned data fools the methods into aligning incorrect classes from the two domains ((b) and (d)). The mismatch between the source and target classes is dependent on the labels of the poison data (due to which the target class 1 is aligned to the source class 2).

performance of our poisoning attacks against other popular UDA methods are present in Appendix F. For all experiments, we train UDA algorithms using neural networks whose architectures are similar to those used by previous works (see Appendix H for details).

## 4.1    Poisoning using mislabeled source and target domain data

In this experiment, mislabeled data is added to the clean source domain as poison. The labels provided by the attacker to the poison data are chosen to fool UDA methods into learning a representation that incurs large errors on the target domain. We use two simple and effective labeling functions for this. For the Digits dataset, we use a labeling function that systematically labels the poison data to the class next to their true class (e.g. poison points with true class one are labeled as two, points with true class two are labeled as three, and so on). For the Office-31 dataset, we use a labeling function that assigns the poison data the label of the closest (in the representation space learned using the clean source domain data) incorrect source domain class. Here the attacker is limited to adding only 10% poisoned data with respect to the size of the available target domain data (for experiments with different poison percentages see Appendix D). In wrong-label correct-domain poisoning, mislabeled source domain data are used as poisons. The results in rows marked with Poison$_{source}$ in Tables 1 and 2, show that UDA methods suffer only a minor decrease in target domain accuracy with this approach. This happens because the presence of a large amount of correctly labeled source domain data prevents the small amount of poisoned data from affecting the performance of UDA methods. A similar effect is observed when a small amount of mislabeled data is used for poisoning in the traditional single domain setting [23, 26]. However, if the relative size of poisoned data is larger or comparable to the size of clean source domain data, poisoning can be effective. This is observed in Table 2 when Amazon is the target data. The Amazon dataset is roughly 5 times bigger than both DSLR and Webcam datasets. Due to this, the permissible amount of poisoned data (10% of the target domain) makes the size of clean and poisoned data comparable, leading to successful poisoning. Thus, this attack causes UDA methods to fail in presence of a large amount of poisoned data.

In the wrong-label wrong-domain approach, mislabeled target domain data is used for poisoning. The effect of poisoning on discriminator-based methods [9, 18] is shown in Fig. 4. The domain discriminator used in these methods is maximally confused when marginal distributions of the source and target domains are aligned. However, alignment of the marginal distributions does not ensure alignment of the conditional distributions [36, 18]. The objective of achieving

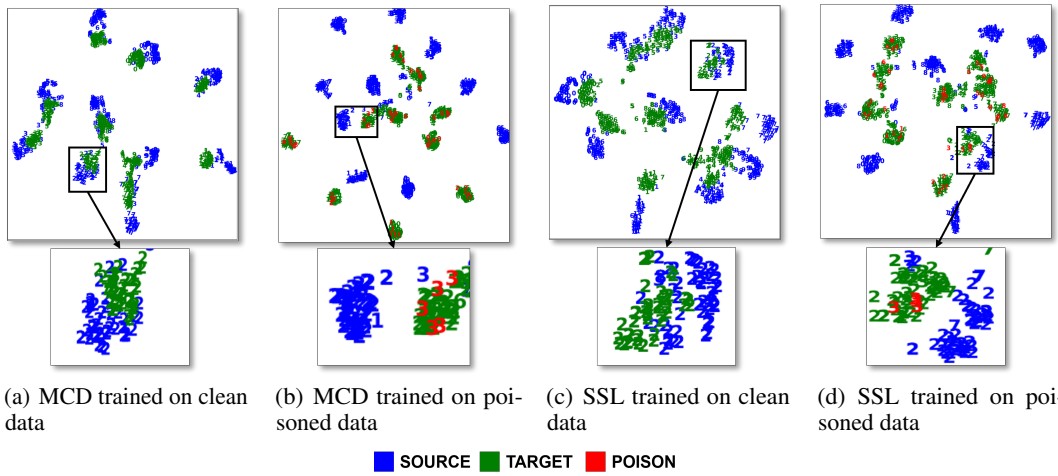

(a) MCD trained on clean data    (b) MCD trained on poisoned data    (c) SSL trained on clean data    (d) SSL trained on poisoned data

🟦 SOURCE    🟩 TARGET    🟥 POISON

Figure 3: (Best viewed in color.) t-SNE embedding of the data in the representation space (for MNIST → USPS task) learned using MCD and SSL on clean and poisoned source domain data. Without poisoning, correct classes (data from source class 2 is zoomed in) from two domains are aligned ((a) and (c)). The presence of poisoned data prevents the methods from aligning correct classes from the two domains ((b) and (d)).

Table 1: Decrease in the target domain accuracy for UDA methods trained on poisoned source domain data (with poisons sampled from source/target domains) compared to accuracy attained with clean data on the Digits tasks (mean±s.d. of 5 trials).

| Method | Data | SVHN → MNIST | MNIST → MNIST_M | MNIST → USPS | USPS → MNIST |
|---|---|---|---|---|---|
| Source only | Clean | 72.42±1.44 | 39.05±2.30 | 87.13±1.75 | 78.6±1.45 |
| DANN | Clean | 78.05±1.15 | 76.22±2.38 | 92.17±0.73 | 92.73±0.71 |
| | Poison$_{source}$ | 70.26±2.84 | 69.98±3.49 | 93.44±0.84 | 92.08±0.68 |
| | Poison$_{target}$ | **1.46±1.12** | **0.48±0.04** | **0.97±0.53** | **5.83±0.82** |
| CDAN | Clean | 79.19±0.70 | 73.88±1.10 | 93.92±0.97 | 95.94±0.71 |
| | Poison$_{source}$ | 73.67±4.19 | 73.36±1.31 | 92.06±0.59 | 92.85±0.31 |
| | Poison$_{target}$ | **12.27±5.02** | **0.59±0.12** | **1.92±0.42** | **2.96±0.71** |
| MCD | Clean | 96.18±1.53 | 93.95±0.33 | 89.96±2.04 | 88.34±2.50 |
| | Poison$_{source}$ | 85.86±5.66 | 93.33±0.71 | 87.99±1.05 | 83.19±2.98 |
| | Poison$_{target}$ | **0.97±0.94** | **0.37±0.06** | **0.66±0.16** | **2.07±0.69** |
| SSL | Clean | 66.85±2.30 | 92.76±0.91 | 88.69±1.28 | 82.23±1.59 |
| | Poison$_{source}$ | 61.97±1.62 | 91.35±1.13 | 85.74±2.92 | 82.56±0.84 |
| | Poison$_{target}$ | **0.31±0.03** | **0.36±0.02** | **7.76±1.52** | **9.88±1.07** |

low source domain error pushes the source domain classifier to correctly classify the poison data. Thus placing the poison and source data with the same labels close in the representation space.

Since the poison data is mislabeled target domain data, poisoning makes UDA methods align wrong source and target domain classes. This leads to a significant decline in the target domain accuracy.

This is also evident from the t-SNE embedding for DANN and CDAN methods in Fig. 2. In absence of poisoning, correct source and target classes are aligned (Fig. 2 (a) and (c)), whereas in presence of poisoning wrong classes from the two domains are closer (Fig. 2 (b) and (d)). For MCD [27], which uses use classifier discrepancy to detect and align source and target domains, our poisoned data prevents the method from detecting target examples. This happens because the term that minimizes the error on the poisoned source domain reduces the discrepancy of the classifiers on poison data, which are from the target domain. Thus, both the generator (common representation) and discriminator (in the form of two classifiers) become optimal and there is no signal for the generator to align the two domains. The t-SNE

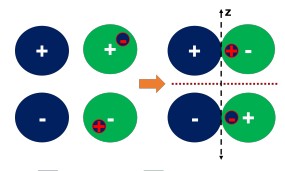

🟦 SOURCE    🟩 TARGET

Figure 4: Wrong-label incorrect-domain poisoning causes discriminator-based UDA approaches to align wrong classes (+ to -) from the two domains, leading to a significant decline in the target domain accuracy.

Table 2: Decrease in the target domain accuracy for UDA methods trained on poisoned source domain data (with poisons sampled from source/target domains) compared to accuracy attained with clean data on the Office tasks (mean±s.d. of 3 trials).

| Method | Dataset | A → D | A → W | D → A | D → W | W → A | W → D |
|---|---|---|---|---|---|---|---|
| Source Only | Clean | 79.61 | 73.18 | 59.33 | 96.31 | 58.75 | 99.68 |
| DANN | Clean | 84.06 | 85.41 | 64.67 | 96.08 | 66.77 | 99.44 |
| | Poison$_{source}$ | 79.11±0.35 | 83.98±1.19 | 44.31±2.94 | 95.22±0.22 | 43.35±1.65 | 96.58±0.87 |
| | Poison$_{target}$ | **59.83±0.20** | **63.18±1.96** | **17.58±0.39** | **76.43±0.62** | **19.82±0.33** | **84.20±0.71** |
| CDAN | Clean | 89.56 | 93.01 | 71.25 | 99.24 | 70.32 | 100 |
| | Poison$_{source}$ | 90.16±0.61 | 90.94±0.13 | 53.68±0.37 | 98.45±0.07 | 57.27±0.57 | 99.66±0.23 |
| | Poison$_{target}$ | **71.88±0.20** | **71.94±0.76** | **11.19±1.47** | **86.37±0.36** | **18.54±0.45** | **89.08±1.23** |
| IW-DAN | Clean | 84.3 | 86.42 | 68.38 | 97.13 | 67.16 | 100 |
| | Poison$_{source}$ | 81.25±0.91 | 83.27±0.45 | 50.76±1.58 | 96.68±0.29 | 48.31±2.02 | 99.73±0.12 |
| | Poison$_{target}$ | **61.64±0.53** | **63.43±1.14** | **15.69±1.76** | **80.29±0.07** | **26.54±0.48** | **88.62±0.23** |
| IW-CDAN | Clean | 88.91 | 93.23 | 71.9 | 99.3 | 70.43 | 100 |
| | Poison$_{source}$ | 89.83±0.31 | 90.77±1.27 | 57.51±0.06 | 98.41±0.07 | 61.16±1.21 | 99.66±0.12 |
| | Poison$_{target}$ | **72.62±0.42** | **70.15±2.21** | **14.36±0.66** | **88.26±0.15** | **22.36±0.96** | **87.75±0.53** |

embedding in Fig. 3 shows this effect. In presence of poisoned data Fig. 3 (b), we see twenty distinct clusters rather than just ten (we have ten classes in the Digits) as seen in the absence of poisoning in Fig. 3 (a). In SSL [35], the generator must work well on the main task, i.e., should correctly classify all data in the poisoned source domain data. The auxiliary task ensures that representations of the source and target domains become similar as seen on clean data (Fig. 3 (c)). But in presence of poisoned data, similar representations of correct source and target domain classes leads to a drop in the accuracy of the main task on the poisoned data. This creates a conflict between the main and auxiliary tasks due to which correct source and target domain classes cannot be aligned (Fig. 3 (d)). The objective of making the main task accurate on poisoned data leads to target domain data being assigned the labels of the poisoned points. Thereby leading to a significant drop in the target-domain accuracy. The results of wrong-label wrong-domain poisoning present in rows marked with Poison$_{target}$ in Tables 1 and 2 show a significant reduction in the target domain accuracy compared to the accuracy obtained on clean data. On Digits, poisoning makes the target domain accuracy close to 0% on most tasks. On Office-31, poisoning causes at least a 20% reduction in the target domain accuracy in most cases. The reason tasks in Office-31 are less hurt by poisoning is because of the use of a pre-trained representation. Similar to previous works for Office-31, we use all labeled source and unlabeled target domain data for training and fine-tune from a representation pre-trained on the ImageNet dataset. The slowly changing representation pre-trained on a massive dataset weakens the effect of poisoning but cannot eliminate it. For the two tasks D → W and W → D, the fine-tuned representation, trained just on the clean source dataset (Source Only in Table 2) achieves high accuracy indicating the domains are already well aligned in terms of conditional distributions as well. As a result, UDA methods can easily align correct classes and suffer a drop close to 10% which is the amount of poisoned data added. However, this is not an interesting case for the evaluation of UDA methods as domains can be aligned just by training on the source domain data without the need for target domain data.

## 4.2 Poisoning with mislabeled watermarked data

In this experiment, we evaluate the effect of using poisoned data that looks like the source domain data. The poisoned data is generated by superimposing an image from the source domain with an image from the target domain. This method of generating poison data is known as watermarking [28]. To generate watermarked poison data we select an image from the target domain ($t$) and a base image from the source domain ($s$) such that it has the same class as the target domain image and lies closest to the target image (in the input space). The poisoned image ($p$) is obtained by a convex combination of the base and target images i.e., $p = \alpha t + (1 - \alpha)s$ where $\alpha \in [0, 1]$. $\alpha$ is selected such that the target image is not visible in the poison image ensuring the poisoned image looks like the image from the source domain. We use the same labeling function as discussed in the previous section to label the poisoned image and add 10% poison data to the source.

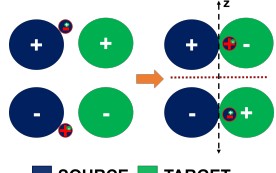

SOURCE TARGET

Figure 5: Successful poisoning with mislabeled watermarked data prevents discriminator-based UDA approaches from aligning correct classes from the source and target domains.

Table 3: Decrease in target accuracy when training different domain adaptation methods on poisoned watermarked data in comparison to the target accuracy obtained with clean data on the Digits task (mean±s.d. of 5 trials).

| Method | Dataset | SVHN → MNIST | MNIST → MNIST_M | MNIST → USPS | USPS → MNIST |
|---|---|---|---|---|---|
| DANN | Clean | $78.05\pm1.15$ | $76.22\pm2.38$ | $92.17\pm0.73$ | $92.73\pm0.71$ |
| | Poisoned$_\alpha$ | $68.76\pm3.91_{0.05}$ $57.96\pm5.84_{0.10}$ $33.33\pm4.38_{0.15}$ | $27.36\pm15.77_{0.05}$ $7.19\pm2.59_{0.10}$ $4.73\pm0.38_{0.15}$ | $91.84\pm0.55_{0.10}$ $85.51\pm3.01_{0.20}$ $39.29\pm1.34_{0.30}$ | $88.93\pm4.36_{0.10}$ $78.29\pm8.52_{0.20}$ $41.52\pm7.43_{0.30}$ |
| CDAN | Clean | $79.19\pm0.70$ | $73.88\pm1.10$ | $93.92\pm0.97$ | $95.94\pm0.71$ |
| | Poisoned$_\alpha$ | $65.77\pm4.82_{0.05}$ $57.57\pm3.11_{0.10}$ $44.83\pm4.09_{0.15}$ | $55.47\pm3.87_{0.05}$ $7.37\pm1.26_{0.10}$ $6.68\pm1.64_{0.15}$ | $92.05\pm0.96_{0.10}$ $86.54\pm2.43_{0.20}$ $88.67\pm0.44_{0.30}$ | $86.53\pm1.55_{0.10}$ $77.39\pm4.84_{0.20}$ $79.54\pm7.02_{0.30}$ |
| MCD | Clean | $96.18\pm1.53$ | $93.95\pm0.33$ | $89.96\pm2.04$ | $88.34\pm2.50$ |
| | Poisoned$_\alpha$ | $74.96\pm3.20_{0.05}$ $35.85\pm3.23_{0.10}$ $17.01\pm1.52_{0.15}$ | $92.18\pm0.78_{0.05}$ $85.38\pm3.57_{0.10}$ $70.34\pm11.49_{0.15}$ | $6.75\pm4.81_{0.10}$ $0.77\pm0.22_{0.20}$ $0.71\pm0.22_{0.30}$ | $30.35\pm2.30_{0.10}$ $11.34\pm0.77_{0.20}$ $3.28\pm0.94_{0.30}$ |
| SSL | Clean | $66.85\pm2.30$ | $92.76\pm0.91$ | $88.69\pm1.28$ | $82.23\pm1.59$ |
| | Poisoned$_\alpha$ | $44.64\pm2.01_{0.05}$ $10.86\pm1.21_{0.10}$ $3.4\pm1.11_{0.15}$ | $53.33\pm13.48_{0.05}$ $26.64\pm10.1_{0.10}$ $12.14\pm4.66_{0.15}$ | $32.38\pm10.77_{0.10}$ $6.12\pm2.13_{0.20}$ $2.42\pm0.41_{0.30}$ | $34.72\pm1.71_{0.10}$ $21.86\pm1.01_{0.20}$ $11.90\pm0.81_{0.30}$ |

The illustrative picture of the effect of poisoning in this scenario is presented in Fig. 5. Successful poisoning, in this case, works just like in the previous experiment i.e., by making the representations of the data from wrong classes in source and target domains similar for DANN/CDAN, by reducing the discrepancy between the classifiers on target data for MCD and inducing a conflict between the supervised and auxiliary task for SSL. The t-SNE embedding showing the effect of poisoning (Fig. 9) and watermarked poison data (Fig. 10) are shown in the Appendix. We evaluate the effectiveness of this method on the Digits dataset for different values of $\alpha$. The results in Table 3 show a significant decrease in the target domain accuracy even with a small watermarking percentage for all methods except CDAN. This is because the success of CDAN is dependent on the correctness of the pseudo-labels on the target domain data (output of the classifier), which are used in the discriminator. Correct pseudo-labels provide CDAN a positive reinforcement to align correct classes from the two domains, leading to a failure of poisoning. However, as we increase the amount of watermarking, the quality of pseudo labels deteriorates. Thus, providing a negative reinforcement to CDAN that causes the alignment of wrong classes from the two domains.

## 4.3 Poisoning using clean-label source and target domain data

In this experiment, correctly labeled data is used for poisoning. To generate clean-label poison data that can affect the performance of UDA methods we must affect the features of the poison data. This requires solving a bilevel optimization problem [11, 23, 24] which we present in the Appendix E. Due to the high computational complexity involved in solving the bilevel problem, we propose to use a simple alternating optimization to demonstrate the feasibility of a clean label poisoning attack against UDA. We use the setting of previous works [11, 28] and consider misclassification of a single target domain test point $(x_{\text{test}}^{\text{target}}, y_{\text{test}}^{\text{target}})$ rather than affecting the accuracy of the entire target domain as done in the previous two experiments. Let $u =$

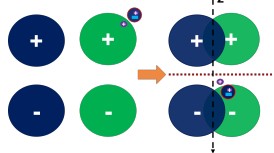

Figure 6: Successful poisoning using clean-label poisoned data aligns the target point (purple) close to the wrong class (-).

$\{u_1, ..., u_n\}$ denote the poisoned data. To ensure a clean label, each poison point $u_i$ must have a bounded perturbation from a base point $x_i^{\text{base}}$ i.e, $\|u_i - x_i^{\text{base}}\| = \|\delta_i\| \le \epsilon$ and has label of the base i.e., $y_i^{\text{base}}$. Thus, $\hat{\mathcal{D}}^{\text{poison}} = \{(u_i, y_i^{\text{base}})\}_{i=1}^{N_{\text{poison}}}$, $\hat{\mathcal{D}}_{\text{source}} = \{(x_i^{\text{source}}, y_i^{\text{source}})\}_{i=1}^{N_{\text{source}}}$ and $\hat{\mathcal{D}}_{\text{target}} = \{(x_i^{\text{target}}, y_i^{\text{target}})\}_{i=1}^{N_{\text{target}}}$. The clean-label poison data $u$ is such that when the victim uses $\hat{\mathcal{D}}^{\text{source}} \bigcup \hat{\mathcal{D}}^{\text{poison}}$ and $\hat{\mathcal{D}}^{\text{target}}$ for UDA, the target domain test point $(x_{\text{test}}^{\text{target}}, y_{\text{test}}^{\text{target}})$ is misclassified.

The optimization problem for the clean-label attack is as follows.

$$\min_u \sum_{i=1}^{N_{\text{poison}}} \|g(x_{\text{test}}^{\text{target}};\theta) - g(u_i;\theta)\|_2^2 \ \text{s.t.} \ \|x_i^{\text{base}} - u_i\| \le \epsilon, \ \text{for i} = 1, ..., N_{\text{poison}}, \tag{4}$$

$$\min_\theta \ \mathcal{L}_{\text{UDA}}(\hat{\mathcal{D}}^{\text{source}} \bigcup \hat{\mathcal{D}}^{\text{poison}}, \hat{\mathcal{D}}^{\text{target}};\theta).$$

The first problem minimizes the distance between the representations of the poison and the target domain test data (first term) while ensuring the poison data is not too far from the base data (second term). The second problem optimizes the parameters of the representation using UDA methods. Attack success is evaluated by solving the second problem in Eq. 4 from scratch and evaluating the classification of $x_{\text{test}}^{\text{target}}$. This is illustrated in Fig. 6. The left part shows the case before retraining using the poison data generated from Eq. 4 and the right part shows how poisoning induces misclassification. We use two approaches for poisoning. The first uses source domain data and the second uses target domain data as base data. We add 1% poisoned data and test the effect of poisoning on a two-class (3 vs 8) domain adaptation problem on MNIST → MNIST_M (see Appendix H). The results in Fig. 7 show that using target domain data as base data is significantly more successful under small permissible perturbation ($\epsilon$). Using base data from the source domain requires larger distortion to keep the poison data close to the target point in the representation space and is hence less successful. This shows the feasibility of clean label attacks against UDA methods. We believe the attack success can be improved by solving the bilevel level problem (Eq. 9 in Appendix E) and is left as future work.

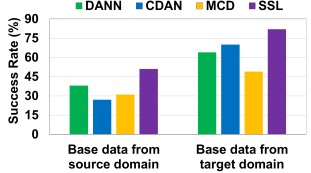

Figure 7: Attack success rate of clean-label poisoning using base data from source/target for a two-class problem in MNIST → MNIST_M.

## 5   Conclusion

We studied the problem of UDA and highlighted the limitations of learning under this setting. We proposed a simple lower bound on the target domain error for UDA, dependent on the labeling function induced by the representation. The lower bound demonstrated that learning a domain invariant representation while minimizing error on the source domain cannot guarantee good generalization on the target domain. We analyzed a simple model and showed the existence of cases where UDA can naturally succeed or fail. The analysis also highlighted a case where, without access to any labeled target domain data, the success and the failure are equally likely. In such a case, the presence of even a small amount of poisoned data can make the data distribution unfavorable for UDA methods, making them fail dramatically in comparison to the case without poisoning. We proposed novel data poisoning attacks to demonstrate the failure of popular UDA methods with a small amount of poisoned data. Our results suggest that the performance of a UDA method in presence of poisoned data indicates how well the method aligns the conditional distributions across the two domains. Thus, we believe our attacks can be used for evaluating UDA methods, beyond simple benchmark datasets, to reveal their robustness to data distributions inherently unfavorable for UDA.

## 6   Acknowledgment

We thank the anonymous reviewers for their insightful comments and suggestions. This work was supported by the NSF EPSCoR-Louisiana Materials Design Alliance (LAMDA) program #OIA-1946231 and by LLNL Laboratory Directed Research and Development project 20-ER-014 (LLNL-CONF-824206). This work was performed under the auspices of the U.S. Department of Energy by the Lawrence Livermore National Laboratory under Contract No. DE-AC52-07NA27344, Lawrence Livermore National Security, LLC [2].

---

[2]This document was prepared as an account of the work sponsored by an agency of the United States Government. Neither the United States Government nor Lawrence Livermore National Security, LLC, nor any of their employees make any warranty, expressed or implied, or assumes any legal liability or responsibility for the accuracy, completeness, or usefulness of any information, apparatus, product, or process disclosed, or represents that its use would not infringe privately owned rights. Reference herein to any specific commercial product,

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
