# Appendix

We present the proof of Theorem 1 in Appendix A followed by the analysis of the illustrative cases of UDA failure in Appendix B. Then we present the results for the experiment of using different poison percentages when mislabeled data is used for poisoning in Appendix D followed by the proposed bilevel formulation for clean label attacks in Appendix E. We discuss additional related work in Appendix G and present additional experiments in Appendix F. We conclude in Appendix H by providing the details of the datasets used, model architectures, and the clean label experiment.

## A  Proof of the lower bound on the target domain loss

**Theorem 1.** *Let $\mathcal{H}$ be the hypothesis class and $\mathcal{G}$ be the class representation maps. Then, for all $h \in \mathcal{H}$ and $g \in \mathcal{G}$,*

$$e_T(h) \geq \max\{e_S(\tilde{f}_S, \tilde{f}_T), e_T(\tilde{f}_S, \tilde{f}_T)\} - e_S(h) - D_1(\tilde{p}_S, \tilde{p}_T).$$

*Proof.*

$$
\begin{aligned}
e_S(h) + e_T(h) &= e_T(h, \tilde{f}_T)) + e_T(h, \tilde{f}_S) + e_S(h, \tilde{f}_S) - e_T(h, \tilde{f}_S) \\
&\geq e_T(\tilde{f}_S, \tilde{f}_T) + e_S(h, \tilde{f}_S) - e_T(h, \tilde{f}_S) \\
&= e_T(\tilde{f}_S, \tilde{f}_T) + \int (\tilde{p}_S(z) - \tilde{p}_T(z))|h(z) - \tilde{f}_S(z)| \, dz \\
&\geq e_T(\tilde{f}_S, \tilde{f}_T) - \int |\tilde{p}_S(z) - \tilde{p}_T(z)| \, dz \\
&= e_T(\tilde{f}_S, \tilde{f}_T) - D_1(\tilde{p}_S, \tilde{p}_T).
\end{aligned}
$$

Similarly, we can also show that

$$e_S(h) + e_T(h) \geq e_S(\tilde{f}_S, \tilde{f}_T) - D_1(\tilde{p}_S, \tilde{p}_T).$$

Combining the two results gives us the statement of the theorem. $\square$

**Corollary 1.2.** *For all $h \in \mathcal{H}$ and $g \in \mathcal{G}$,*

$$|e_T(h) - e_S(\tilde{f}_S, \tilde{f}_T)| \leq e_S(h) + D_1(\tilde{p}_S, \tilde{p}_T), \quad \text{and} \quad |e_T(h) - e_T(\tilde{f}_S, \tilde{f}_T)| \leq e_S(h) + D_1(\tilde{p}_S, \tilde{p}_T).$$

*Proof.* From the upper bound we have,

$$e_T(h) - e_S(\tilde{f}_S, \tilde{f}_T)) \leq e_S(h) + D_1(\tilde{p}_S, \tilde{p}_T) \quad \text{and} \quad e_T(h) - e_T(\tilde{f}_S, \tilde{f}_T)) \leq e_S(h) + D_1(\tilde{p}_S, \tilde{p}_T)$$

From the lower bound (Eq. 2) we have,

$$e_T(h) - e_S(\tilde{f}_S, \tilde{f}_T) \geq -e_S(h) - D_1(\tilde{p}_S, \tilde{p}_T) \quad \text{and} \quad e_T(h) - e_T(\tilde{f}_S, \tilde{f}_T) \geq -e_S(h) - D_1(\tilde{p}_S, \tilde{p}_T)$$

Combining the results from the upper and the lower bounds gives us the statement of the corollary. $\square$

## B  Illustrative examples of UDA failure

In this section, we provide the details of the analysis of the illustrative cases in the main paper. As described in Sec. 3.2, the input space $\mathcal{X}$ is in $\mathbb{R}^2$ and the source and the target distributions are Gaussian mixtures

$$p_S(x) = 0.5p_{S+}(x) + 0.5p_{S-}(x) \quad \text{and} \quad p_T(x) = 0.5p_{T+}(x) + 0.5p_{T-}(x),$$

where $p_{S+}(x) = \mathcal{N}(\mu_{S+}, \sigma^2 I)$, $p_{S-}(x) = \mathcal{N}(\mu_{S-}, \sigma^2 I)$, $p_{T+}(x) = \mathcal{N}(\mu_{T+}, \sigma^2 I)$, and $p_{T-}(x) = \mathcal{N}(\mu_{T-}, \sigma^2 I)$. The true labeling function $f(x)$ in the input space is assumed linear: $f(x) = I[v^T x > 0]$ where $v$ is the unit normal vector to the decision boundary. The representation space $\mathcal{Z}$ is in $\mathbb{R}$ and the representation map $g : \mathcal{X} \to \mathcal{Z}$ is linear: $g(x) = u^T x$ where $\|u\| = 1$. For the hypothesis, we use $h(z) = \Phi(az + b)$ which is a linear model $az + b$ followed by a saturating function which can be the cumulative normal distribution $\Phi$ (or others such as the logistic function $l$).

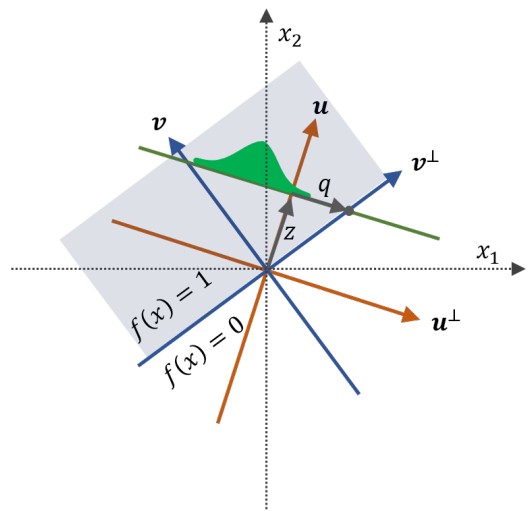

Figure 8: This figure provides a visual help for deriving the induced labeling function $\tilde{f}(z)$ in Eq. 5. $u$ is the direction of the 1-D projection $g(z) = u^T x$, $v$ is the direction of the labeling function $f(x)$ in the input space where we assumed $f(x) = I[v^T x > 0]$, and $q$ is the intersection of the two lines $v^T x = 0$ and $u^T(x - zu) = 0$ projected along the $u^\perp$ direction. Evaluating Eq. 6 using the help of this figure results in Eq. 7.

The representation map $g$ induces the distributions $\tilde{p}(z)$ over $\mathcal{Z}$ as

$$\tilde{p}_S(z) = 0.5\mathcal{N}(u^T\mu_{S+}, \sigma^2) + 0.5\mathcal{N}(u^T\mu_{S-}, \sigma^2), \text{ and}$$

$$\tilde{p}_T(z) = 0.5\mathcal{N}(u^T\mu_{T+}, \sigma^2) + 0.5\mathcal{N}(u^T\mu_{T-}, \sigma^2).$$

The map $g$ also induces the labeling function $\tilde{f}(z)$ on $\mathcal{Z}$ defined as $\tilde{f}(z) = E_{\mathcal{D}}[f(x)|g(x) = z]$ [2]. Computing this quantity can be complex in general but is relatively straightforward for a mixture of Gaussians and a simple half-space labeling function $f(x)$. Following the definition, we have

$$\tilde{f}(z) = E_{\mathcal{D}}[f(x)|g(x) = z] = \int_{\mathcal{Z}} f(x) \, I[u^T x = z] \, p(x|z = g(x))dx. \tag{5}$$

In our example, the integral $\int_{\mathbb{R}^2} \cdot \, dx$ can be decomposed into $\int_{-\infty}^{\infty} \int_{-\infty}^{\infty} \cdot \, dzdw$ where $z$ and $w$ are the coordinates along the rotated axes $u$ and $u^\perp$ (see Fig. 8).

We therefore have

$$\tilde{f}(z) = \int_{-\infty}^{\infty} \int_{-\infty}^{\infty} I[v^T x > 0] \, I[u^T x = z] \, p(x|z = g(x))dzdw$$

$$= \int_{-\infty}^{\infty} I[v^T x > 0] \, (0.5\mathcal{N}_w(\mu_+^T u^\perp, \sigma^2 I) + 0.5\mathcal{N}_w(\mu_+^T u^\perp, \sigma^2 I))dw. \tag{6}$$

This integral can be evaluated as

$$\tilde{f}(z) = \begin{cases} 0.5\Phi(\frac{q-\mu_+^T u^\perp}{\sigma}) + 0.5\Phi(\frac{q-\mu_+^T u^\perp}{\sigma}) & \text{if } v^T u^\perp < 0 \\ 0.5[1 - \Phi(\frac{q-\mu_+^T u^\perp}{\sigma})] + 0.5[1 - \Phi(\frac{q-\mu_+^T u^\perp}{\sigma})] & \text{if } v^T u^\perp > 0 \\ 0.5(1 + sign(z)) & \text{if } v^T u^\perp = 0 \text{ and } v^T u > 0 \\ 0.5(1 - sign(z)) & \text{if } v^T u^\perp = 0 \text{ and } v^T u < 0 \end{cases} \tag{7}$$

where $\Phi$ is the cumulative normal distribution and $q$ is the intersection of the two lines $v^T x = 0$ and $u^T(x - zu) = 0$ projected along the $u^\perp$ direction. More concretely,

$$q = \frac{u_1 v_1 + u_2 v_2}{u_1 v_2 - u_2 v_1} z$$

where $u = [u_1, u_2]^T$ and $v = [v_1, v_2]^T$.

The UDA minimization problem is

$$\min_{u,a,b} \ e_S(h) + \lambda D(\tilde{p}_S, \tilde{p}_T) + \eta(\|u\|^2 - 1)^2, \tag{8}$$

where the last term was added to enforce $\|u\| = 1$. For differentiability, we consider the squared loss instead of the absolute loss:

$$e_S(h) = E_S[(\Phi(az + b) - \tilde{f}_S(z))^2] = \int_{\mathbb{R}} \tilde{p}_S(z) \left( \Phi(az + b) - \tilde{f}_S(z) \right)^2 dz$$

and also

$$D(\tilde{p}, \tilde{p}') = \int_{\mathbb{R}} (\tilde{p}(z) - \tilde{p}'(z))^2 dz.$$

The expectation in $e_S(h)$ can only be computed numerically due to the complex formula for $\tilde{f}(z)$. On the other hand, the mismatch loss is

$$
\begin{aligned}
D(\tilde{p}_S(z), \tilde{p}_T(z)) &= \int_{\mathbb{R}} (\tilde{p}_S(z) - \tilde{p}_T(z))^2 dz \\
&= \int_{\mathbb{R}} \left( \frac{0.5}{2\pi\sigma^2} \right)^2 \left[ e^{-\frac{(z - u^T \mu_{S+})^2}{2\sigma^2}} + e^{-\frac{(z - u^T \mu_{S-})^2}{2\sigma^2}} - e^{-\frac{(z - u^T \mu_{T+})^2}{2\sigma^2}} - e^{-\frac{(z - u^T \mu_{T-})^2}{2\sigma^2}} \right]^2 dz,
\end{aligned}
$$

which can be computed either numerically or analytically.

The three cases explained in the main paper are as follows:

Case 1 : $\mu_{S+} = [-1, 1]^T$, $\mu_{S-} = [-1, -1]^T$, $\mu_{T+} = [1, 1]^T$, $\mu_{T-} = [1, -1]^T$, $v_S(x) = v_T(x) = [0, 1]^T$, $\lambda = 10^{-1}$

Case 2 : $\mu_{S+} = [-1, 1]^T$, $\mu_{S-} = [-1, -1]^T$, $\mu_{T+} = [1, -1]^T$, $\mu_{T-} = [1, 1]^T$, $v_S(x) = -v_T(x) = [0, 1]^T$, $\lambda = 10^{-1}$

Case 3 : $\mu_{S+} = [0, 1]^T$, $\mu_{S-} = [0, -1]^T$, $\mu_{T+} = [-1, 0]^T$, $\mu_{T-} = [1, 0]^T$, $v_S(x) = [0, 1]^T$, $v_T(x) = [-1, 0]^T$, $\lambda = 10^{-2}$

The other shared parameters are $\sigma = 1$ and $\eta = 10$. The $\lambda$ determines the optimal tradeoff between $E_s$ and $D$ in Eq. 8.

We solve Eq. 8 numerically using *scipy.optimize.minimize(method='Nelder-Mead')* function which is stable even if the cost function may be non-differentiable. Starting from random initial conditions and running until convergence, the solution $u$ for both Case 1 and Case 2 converges to $[0, 1]^T$.

For Case 1 (favorable case), we get $\max\{e_S(\tilde{f}_S, \tilde{f}_T), e_T(\tilde{f}_S, \tilde{f}_T)\} < 10^{-3}$ and $e_T(h) < 10^{-3}$ which shows UDA was successful.

For Case 2 (unfavorable case), we get $\max\{e_S(\tilde{f}_S, \tilde{f}_T), e_T(\tilde{f}_S, \tilde{f}_T)\} > 0.99$ and $e_T(h) > 0.99$ which shows UDA was unsuccessful.

For Case 3 (ambiguous case), there is an almost equal chance of $u$ converging to $[-0.70, 0.72]^T$ or $[0.70, 0.72]^T$. For the former, we get $\max\{e_S(\tilde{f}_S, \tilde{f}_T), e_S(\tilde{f}_S, \tilde{f}_T)\} < 10^{-4}$ and $e_T(h) < 10^{-3}$ where UDA is successful. For the latter, we get $\max\{e_S(\tilde{f}_S, \tilde{f}_T), e_S(\tilde{f}_S, \tilde{f}_T)\} \cong 0.33$ and $e_T(h) \cong 0.33$ where UDA has failed.

## C   Details of the figures explaining the effect of poisoning on UDA methods

As illustrated in Case 3 of Fig. 1 in Sec. 3.2, UDA methods can be fooled into producing a representation that causes a large error on the target domain with a small amount of poisoned data. The simplest successful poisoning attack to fool UDA methods was shown in Sec. 4.1 (wrong-label incorrect-domain poisoning). In this attack, we added mislabeled data (wrong-label) from the target domain into the source data (incorrect-domain). The left part of Fig. 4 shows this setting. The right part of Fig. 4 shows how the representation learned from discriminator-based UDA methods aligns the incorrect classes closer than the correct ones. Due to the lack of target domain labels, the loss of the discriminator is minimized as long as the green and blue blobs align, regardless of their labels. But to minimize the classification loss on the source domain the representation must classify the

Table 4: Effect of using different percentages of wrong-label incorrect-domain poisoned data on the target domain accuracy when training UDA methods on poisoned source domain data on the Digits tasks (mean±s.d. of 5 trials).

| Poison$_{target}$ (%) | DANN | | CDAN | | MCD | | SSL | |
|---|---|---|---|---|---|---|---|---|
| | MNIST → USPS | USPS → MNIST | MNIST → USPS | USPS → MNIST | MNIST → USPS | USPS → MNIST | MNIST → USPS | USPS → MNIST |
| 0% (Clean) | 92.17±0.73 | 92.73±0.71 | 93.92±0.97 | 95.94±0.71 | 89.96±2.04 | 88.34±2.50 | 88.69±1.28 | 82.23±1.59 |
| 2% | 63.53±2.09 | 94.72±0.63 | 90.54±0.91 | 88.79±2.34 | 22.74±2.17 | 51.02±3.57 | 65.88±2.93 | 41.25±2.32 |
| 4% | 28.39±4.78 | 34.25±9.53 | 90.22±0.74 | 76.55±2.25 | 2.37±1.41 | 16.66±4.73 | 30.82±1.28 | 28.60±2.16 |
| 6% | 7.32±4.78 | 12.96±7.33 | 42.86±5.09 | 8.61±4.77 | 2.56±0.97 | 4.64±1.34 | 21.29±2.51 | 18.89±1.11 |
| 8% | 0.97±0.44 | 1.63±0.41 | 7.02±3.88 | 5.35±0.94 | 7.04±0.25 | 4.43±1.76 | 10.84±1.52 | 11.11±2.74 |
| 10% | 0.97±0.53 | 5.83±0.82 | 1.92±0.42 | 2.96±0.71 | 0.66±0.16 | 2.07±0.69 | 7.76±1.52 | 9.88±1.07 |

poison data correctly. This forces the representation of wrong source and target domain classes to be closer than the correct ones. As a result of this, the source classification loss and domain mismatch loss are minimized but the learned representation still incurs a large target domain error. This is exactly what Case 3 (right) of Fig. 1 illustrated. The t-SNE embeddings in Fig. 2 confirm this on real datasets where representations are learned using popular discriminator-based UDA methods.

To make the poisoning attacks harder to detect we used watermarking-based attacks using poison data that has some features of the target data but still looks like the source data (Fig. 10). This setting is illustrated in the left part of Fig. 5. The right part of Fig. 5 shows how discriminator-based UDA methods are fooled into producing a representation that fails to generalize on the target domain. Similar to the previous case of wrong-label incorrect-domain poisoning discriminator is optimal when the green and blue blobs align. However, since the poison data has incorrect labels the source classification loss prefers to align it with the wrong class (poison data labeled as + is aligned with source class with label +). Due to the presence of target features in the poison data (due to watermarking) the representation moves the target domain data closer to the poison data leading to an alignment of wrong source and target domain classes. As the percentage of watermarking increases the poisoning attack becomes more successful (Table 3) indicating that target domain data is being aligned to wrong source domain classes similar to the poison data. Fig. 9 ((a) and (b)) demonstrate this effect on popular discriminator-based UDA methods.

To make our attack even stealthier, we consider the effect of using correctly labeled poisoned data on the UDA methods. We generate such poison data by solving a computationally efficient version of the bilevel problem (Eq. 9), as shown in Eq. 4. The left part of Fig. 6 illustrates the poison data generated by solving Eq. 4. The poison data specifically targets a particular target domain test point (shown in purple). Since the poison data is close in the representation space to the target domain test point, UDA methods align it closer to the class of the poison data. This leads to misclassification of the test point. Since the poison data is crafted for a specific test point, they don't have much effect on the entire target domain data. The right part of Fig. 6 illustrates this effect and shows that clean labeled poison data can also successfully hurt the performance of UDA methods.

## D   Effect of poison percentage on attack success with mislabeled poison data

In this section, we evaluate the effect of using different poison percentages on attack success when mislabeled data is used for poisoning. As can be seen in Tables 1 and 2, the success of wrong-label clean-domain poisoning with 10% poisoned data is very limited. Thus, here we only focus on using a smaller poison percentage to study the attack success of wrong-label wrong-domain poisoning. The results of the experiment are summarized in Table 4. For all tasks, the presence of only 6% poison data causes a significant decrease in the target domain accuracy. When the poison percentage is decreased further to only 2% we still see a drop of at least 20% in the target accuracy for all methods except CDAN[18]. The use of a conditional discriminator provides CDAN this robustness. However, the success of CDAN is dependent on the quality of the pseudo-labels from the classifier on the target domain data. Good pseudo-labels provide CDAN a positive reinforcement to align correct source and target domain classes. Thus, leading to a failure of poisoning. However, as the percentage of poisoned data increases, the classifier begins to easily classify the target domain data into labels intended by the attacker, deteriorating the quality of the pseudo-labels. This provides a negative reinforcement to CDAN causing it to align wrong classes from the source and the target domain. As a result, the poisoning attack becomes successful. Thus, for wrong-label wrong-domain poisoning, increasing the percentage of poison data gradually drives UDA methods from the case favorable to UDA to the unfavorable one.

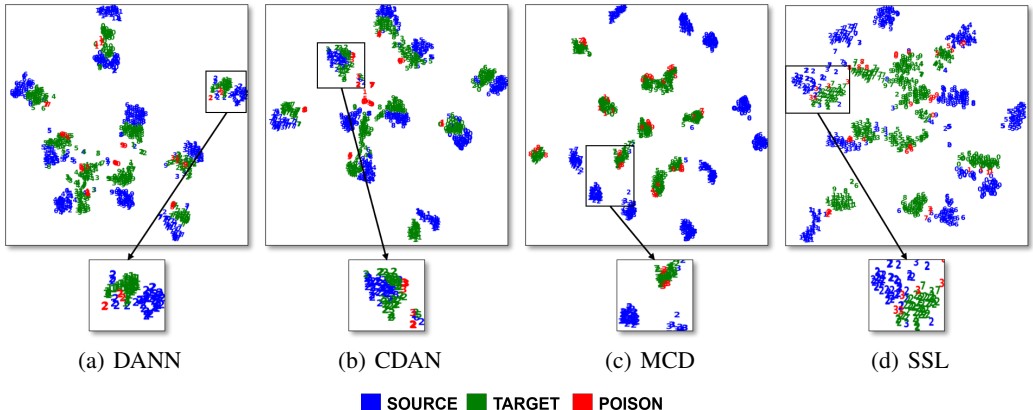

■ SOURCE ■ TARGET ■ POISON

(a) DANN      (b) CDAN      (c) MCD      (d) SSL

Figure 9: (Best viewed in color). t-SNE embedding of the data in the representation space (for MNIST $\rightarrow$ USPS task) learned using DANN, CDAN, MCD, and SSL on source domain data poisoned with watermarked ($\alpha = 0.3$) data. Successful poisoning aligns the wrong classes for discriminator-based approaches, as seen in (a) with DANN. Poisoning fails against CDAN because of the pseudo-labels being correct on the target data (b). For MCD, we see 20 distinct clusters highlighting the failure of the method at detecting and aligning target domain data (c). For SSL, the poison data has prevented the correct classes from having very similar representations (d). The failure of most UDA methods with a small amount of watermarked data makes our attack practical and raises serious concerns about the success of these methods.

## E  Bilevel formulation for clean-label attacks

In this section, we present the bilevel formulation for a clean-label data poisoning attack against UDA methods. Let $u = \{u_1, ..., u_n\}$ denote the poisoned data and $\hat{\mathcal{D}}^{\text{val}_{\text{target}}} = \{(x_i^{\text{val}_{\text{target}}}, y_i^{\text{val}_{\text{target}}})\}_{i=1}^{N_{\text{val}_{\text{target}}}}$, be a small set of labeled target domain data accessible to the attacker. To ensure a clean label, each poison point $u_i$ must have a bounded perturbation from a base point $x_i^{\text{base}}$ i.e, $\|u_i - x_i^{\text{base}}\| = \|\delta_i\| \leq \epsilon$ and has label of the base i.e., $y_i^{\text{base}}$. Thus, $\hat{\mathcal{D}}^{\text{poison}} = \{(u_i, y_i^{\text{base}})\}_{i=1}^{N_{\text{poison}}}$. The clean-label poison data $u$ is such that when the victim uses $\hat{\mathcal{D}}^{\text{source}} \bigcup \hat{\mathcal{D}}^{\text{poison}}$ and $\hat{\mathcal{D}}^{\text{target}}$ for UDA, the accuracy on $\hat{\mathcal{D}}^{\text{val}_{\text{target}}}$ is minimized. The bilevel formulation for this attack is as follows:

$$\max_{u \in \mathcal{U}} \ \mathcal{L}(\hat{\mathcal{D}}^{\text{val}_{\text{target}}}; \theta^*) \ \text{s.t.} \ \theta^* = \arg\min_{\theta} \ \mathcal{L}_{\text{UDA}}(\hat{\mathcal{D}}^{\text{clean}} \bigcup \hat{\mathcal{D}}^{\text{poison}}, \hat{\mathcal{D}}^{\text{target}}; \theta). \qquad (9)$$

The solution to the lower-level problem $\theta^*$ are the parameters of the generator and the classifier learned from using a UDA method on the poisoned source domain data and unlabeled target domain data. Solving bilevel optimization problems [11, 23, 24] to generate clean-label poison data has previously been shown to be effective. We used an alternating optimization to avoid the computational complexity of solving the bilevel optimization (Eq. 4). However, we believe the attack success can be boosted by solving the bilevel formulation proposed in Eq. 9 and is left for future work.

## F  Additional experiments

### F.1  Effect of changing the percentage of poison data and divergence between the domains

We use a simple two moons dataset to illustrate the effect of changing both the poison percentage and the divergence between the two domains. The target domain data is the translated version of the source domain data along the x-axis. The amount of translation is used to measure the divergence between the source and the target domains. For this experiment, we use mislabeled target domain data as our poison points. This is referred to as wrong-label incorrect-domain poisoning in the main paper. The results of this experiment are present in Table 5. Consistent with the results present in rows marked with Poison_target of Table 1 and 2 of the main paper and Table 4, we see that increasing the divergence between the source and target domains makes it easy to poison UDA methods with small

Table 5: Effect of using different percentages of wrong-label incorrect-domain poisoned data on the target domain accuracy when the divergence (D) between the source and target domain is changed. (mean±s.d. of 5 trials).

| Method | Dataset | Target domain accuracy(%) | | |
| --- | --- | --- | --- | --- |
| | | D=0.25 | D=0.5 | D=0.75 |
| Source Only | Clean | 99.60±0.01 | 87.92±0.96 | 68.4±0.13 |
| | Poisoned (5%) | 79.76±2.94 | 55.24±4.42 | 59.44±2.77 |
| | Poisoned (10%) | 53.16±2.44 | 46.52±3.51 | 42.56±2.65 |
| DANN | Clean | 98.48±1.45 | 97.24±1.27 | 71.80±3.92 |
| | Poisoned (5%) | 95.36±2.54 | 83.08±1.43 | 62.72±3.63 |
| | Poisoned (10%) | 85.52±10.1 | 65.56±8.44 | 40.92±5.97 |
| CDAN | Clean | 100±0.0 | 94.52±1.31 | 76.08±4.49 |
| | Poisoned (5%) | 91.36±2.98 | 77.48±6.23 | 64.28±4.95 |
| | Poisoned (10%) | 87.12±1.35 | 73.16±0.81 | 46.04±8.73 |
| MCD | Clean | 100±0.0 | 91.88±2.35 | 69.28±2.77 |
| | Poisoned (5%) | 79.68±5.52 | 70.48±10.69 | 58.88±9.96 |
| | Poisoned (10%) | 65.81±2.66 | 44.28±5.73 | 40.96±16.4 |

Table 6: Decrease in the target domain accuracy for DAN trained on poisoned source domain data (with poisons sampled from the target domain) compared to accuracy attained with clean data on the Office tasks (mean±s.d. of 3 trials).

| Dataset | A → D | A → W | D → A | D → W | W → A | W → D |
| --- | --- | --- | --- | --- | --- | --- |
| Clean | 82.3 | 80.1 | 68.9 | 98.0 | 66.1 | 99.0 |
| Poisoned | 59.8 | 51.4 | 6.7 | 53.7 | 7.6 | 79.9 |

amounts of poisoned data. Moreover, at a fixed divergence level increases the amount of poisoned data leads to a larger drop in the target domain accuracy.

**Effect of poisoning on [17]:** Following the original and follow-up works on MMD [6], we also evaluated the effect of our poisoning attacks against this method using the Office-31 dataset. The results of poisoning on DAN (Table 6) with 10% mislabeled target domain data, consistent with the other results shown in Table 2 of the main paper, demonstrate the effectiveness of our poisoning attack against this method. We used the Pytorch version of the implementation for DAN available from the official code [6] to generate these results.

**Effect of poisoning on [30]:** Here we present the results of our poisoning attack against the DIRT-T method proposed by [30], using the Digits dataset. This method uses virtual adversarial training and conditional entropy minimization whose effectiveness is contingent on the quality of the pseudo-labels of the target domain data. In the main paper, we presented results of using a similar method, CDAN[18], which relied on the idea of using pseudo-labels for the target domain data in the discriminator. Consistent with our results of CDAN presented in Table 1, we see our poisoning attacks (Table 7) are effective at reducing the target domain accuracy obtainable by using the DIRT-T method. We used 10% mislabeled target domain data as our poisons (same as that used in Table 1) and used a similar setting to the official implementation of the paper for our evaluation.

Table 7: Decrease in the target domain accuracy for DIRT-T trained on poisoned source domain data (with poisons sampled from the target domain) compared to accuracy attained with clean data on the Digits tasks (mean±s.d. of 5 trials).

| Data | SVHN → MNIST | MNIST → MNIST_M | MNIST → USPS | USPS → MNIST |
| --- | --- | --- | --- | --- |
| Clean | 92.45±0.46 | 91.41±0.16 | 98.15±0.28 | 98.13±0.09 |
| Poisoned | 0.09±0.02 | 0.37±0.01 | 4.62±4.33 | 0.31±0.24 |

Table 8: Decrease in the target domain accuracy for CYCADA trained on poisoned source domain data (with poisons sampled from the target domain) compared to accuracy attained with clean data on the Digits tasks (mean±s.d. of 5 trials).

| Method | SVHN $\rightarrow$ MNIST | MNIST $\rightarrow$ USPS | USPS $\rightarrow$ MNIST |
|---|---|---|---|
| Transformed Source Only (Clean data) | 74.5±0.3 | 95.6±0.2 | 96.4±0.1 |
| Feature level (Clean data) | 90.4±0.4 | 95.6±0.2 | 96.5±0.1 |
| Feature level (Poisoned data) | 84.3±2.3 | 93.3±0.5 | 60.7±3.5 |

**Effect of poisoning on [10]:** This work uses a Cycle GAN coupled with a semantic loss dependent on the pseudo-labels for the target domain data, to enforce consistency of the two domains in the pixel space as well as in the representation space. When using the poisoned data to train the cycle GAN, we found that it failed to generate good transformations of the source domain. However, it could not be concluded whether the failure of cycle GAN was due to poisoning or due to hyperparameters. Thus, in our poisoning experiments, we omit the training of the cycle GAN and rely on the transformed version of the source domain images provided in the official repository. We treat these transformed images as our original source domain images. We present the results of poisoning only the feature level domain adaptation in Table 8 (assuming cycle GAN is trained on clean data). The high target accuracy of training using only the transformed source domain data (transformed source only) compared to the target accuracy obtained using feature-level domain adaptation on clean data implies a high similarity between the transformed source and the target domains. Due to this high similarity between the transformed source and the target domains, our poisoning attacks (using 10% mislabeled target domain data as poisoned data) have limited effect. For USPS $\rightarrow$ MNIST, poisoning with 10% of MNIST training data (6000 images) is comparable to the size of USPS training data (7291 images) and thus we see a significant reduction in the target domain accuracy. These results are consistent with the results presented in the main paper, especially on tasks W $\rightarrow$ D and D $\rightarrow$ W in Table 2, where poisoning fails since the source and target domains are very similar (indicated by high source only performance).

# G  Additional related work

Previous work [36], presented an information-theoretic lower bound to explain the failure of learning a domain invariant representation when marginal label distributions differ across the source and target domains. In comparison, our lower bound does not require any assumption on the data distributions. Our bound presents a necessary condition for successful learning in the UDA setting. In particular, our lower bound shows that a UDA method may succeed or fail at target generalization i.e. the term $max\ (e_S(\tilde{f}_S, \tilde{f}_T),\ e_T(\tilde{f}_S, \tilde{f}_T))$ in our lower bound (Eq. 2) can be small or large, even when a domain invariant representation is learnt that minimizes source error.

Unlike the bounds presented by previous works [36, 14] which fail to provide insights into the behavior of UDA when marginal label distributions are the same across the two domains, our bound remains tight. In particular, the information-theoretic lower bound of [36] (Theorem 4.3 of [36]) becomes vacuous when $d_{JS}(p_S^Y, p_T^Y) = 0$. However, our lower bound will be non-vacuous in this scenario as long as UDA methods are used i.e., $e_S$ and $D_1(\tilde{p}_S, \tilde{p}_T)$ are minimized. A concrete example where our bound is tighter than the bound of Theorem 4.3 of [36] is their example in section 4.1 where the true target risk is 1. As discussed in their paper (last paragraph below Corollary 4.1), their lower bound on the example is vacuous and says that the target risk will be greater than 0. In comparison, our lower bound in Theorem 1 says that the target risk is greater or equal to 1 which is tighter and much more informative.

Compared to the previous work of [34] which decomposed the target risk into source risk, representation conditional label divergence, and representation covariate shift and explained the reason for the failure of DANN to be its inability to account for representation conditional label divergence. Our lower bound suggests a similar explanation for the failure of DANN i.e, without access to any labeled target domain data, DANN may learn a representation that induces labeling functions $(\tilde{f}_S)$ and $(\tilde{f}_T$ which do not agree with each other on the source and target domains. This difference in the induced labeling functions is captured by $max(e_S(\tilde{f}_S, \tilde{f}_T),\ e_T(\tilde{f}_S, \tilde{f}_T))$ term in our lower bound. Our explanation for UDA failure also extends beyond DANN as we illustrated through our data

poisoning attacks, which are concrete ways to lead UDA algorithms to learn a representation that incurs high error on the target domain.

# H  Details of the experiments

All codes are written in Python using Tensorflow/Keras and were run on Intel Xeon(R) W-2123 CPU with 64 GB of RAM and dual NVIDIA TITAN RTX. Dataset details and model architectures used are described below.

## H.1  Dataset description

Here we describe the details of the datasets used for the Digits and Office-31 tasks.

**Digits:** For this task, we use 4 datasets: MNIST, MNIST_M, SVHN, and USPS. We evaluate four popular tasks under this, namely, SVHN $\rightarrow$ MNIST, MNIST $\rightarrow$ MNIST_M, MNIST $\rightarrow$ USPS and USPS $\rightarrow$ MNIST. For SVHN $\rightarrow$ MNIST, we train on 73,257 images from SVHN and 60,000 images from MNIST while testing on 10,000 MNIST images. For MNIST $\rightarrow$ MNIST_M, we use 60,000 from MNIST and MNIST_M for training and test on 10,000 MNIST images. Lastly, for MNIST $\rightarrow$ USPS and USPS $\rightarrow$ MNIST, we use 2,000 images from MNIST and 1,800 images from USPS for training. We test on the 10,000 MNIST images and 1,860 USPS images.

**Office-31:** The dataset contains a total of 4110 images belonging to 31 categories from 3 domains: Amazon (A), DSLR(D), and Webcam(W). We evaluate the performance of UDA on all six tasks, namely, A $\rightarrow$ D, A $\rightarrow$ W, D $\rightarrow$ A, D $\rightarrow$ W, W $\rightarrow$ A, W $\rightarrow$ D.

## H.2  Model architecture

Here we describe the model architectures used for different tasks. To fairly compare the performance of different UDA methods and eliminate the effect of architecture changes in improving the performance of different methods, we make use of similar model architectures for different methods, as described below. The effectiveness of these architectures has also been shown by previous works.

**Digits:** The architectures used for MNIST $\rightarrow$ MNIST_M, MNIST $\rightarrow$ USPS and USPS $\rightarrow$ MNIST involves a shared convolution neural network. The output of this shared network is fed into a softmax classifier and the discriminator. The architecture of the shared network consists of a convolution layer with a kernel size of 5x5, 20 filters, and ReLU activation, followed by a max-pooling layer of size 2x2. This is followed by another convolution layer with a 5x5 kernel, 50 filters, and ReLU activation followed by similar max pooling and a dropout. Then we have a fully connected layer with ReLU activation of size 500 followed by a dropout layer. For the discriminator, we use two dense layers with 500 units each followed by a ReLU and a dropout layer. This is followed by a 2 unit softmax layer. For MCD, we use the following architecture for the generator on MNIST $\rightarrow$ MNIST_M task. A convolution layer with a kernel size of 5x5, 32 filters, and ReLU activation, followed by a max-pooling layer of size 2x2. This is followed by another convolution layer with a 5x5 kernel, 48 filters, and ReLU activation followed by a similar max-pooling layer. We use 2 dense layers for the classifier with 100 units followed by ReLU activation and dropout layers. This is followed by the softmax layer. Unlike the original work MCD[27], we do not use batch normalization layers in these tasks to make architectures consistent across different methods.

For SVHN $\rightarrow$ MNIST we use the following architecture for the generator. A convolution layer with a kernel size of 5x5, 64 filters, the stride of 2 followed by batch normalization, dropout, and ReLU activation layer. This is followed by another convolution layer with a kernel size of 5x5, 128 filters, the stride of 2 followed by batch normalization, dropout, and ReLU activation layer. Then another convolution layer with a kernel size of 5x5, 256 filters, the stride of 2 followed by batch normalization, dropout, and ReLU activation layer. This is followed by a dense layer with 512 units followed by batch normalization, ReLU activation, and a dropout layer. We use the softmax

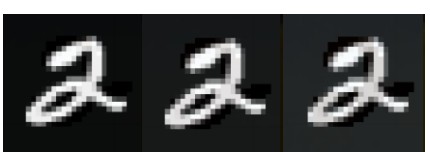

Figure 10: Watermarked poison data for MNIST $\rightarrow$ MNIST_M task with $\alpha$ in $\{0.05, 0.10, 0.15\}$.

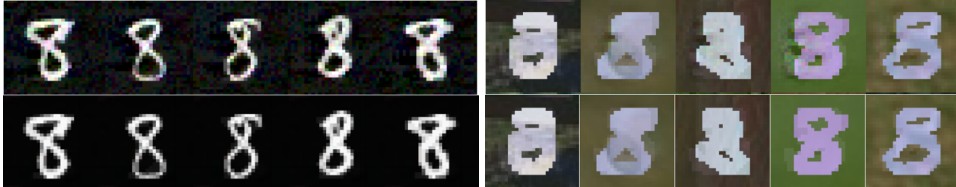

(a) Base data chosen from the source domain      (b) Base data chosen from the target domain

Figure 11: Poison data (top rows) obtained after solving Eq. 4 by using DANN as the UDA method, with base data (bottom rows) initialized from the source domain (left) and the target domain (right). Attack success with poison data initialized from the target is significantly higher than the attack success obtained with poison data initialized from the source, from under the same maximum permissible distortion constraint ($\epsilon = 0.1$ in $\ell_\infty$ norm) as seen in Fig. 7.

layer for classification. For the discriminator, we use two dense layers with 500 units each followed by a ReLU and a dropout layer. This is followed by a 2 unit softmax layer. For MCD, we use the same architecture for the generator except that we use max-pooling instead of convolution layers with stride 2 to downsample the representation. The classifier uses the output of the generator and feeds into a dense layer with 256 units followed by batch normalization and ReLU activation layers. This is followed by a softmax layer.

**Office-31:** For office experiments, we use the publicly available code of the work[3] [6] and supply the poisoned data by adding them to the input files being used by the code. We use all default options of the code and use DAN, CDAN, IW-DAN, IW-CDAN algorithms. This is done to eliminate the effect of hyperparameters on the performance of the UDA algorithms on the Office-31 dataset and be able to fairly compare the performance of poisoning. To obtain the representation trained only on the source domain data, we initialize a ResNet50 model with weight pre-trained on Imagenet. We then update the representation by training on respective source domain data for different tasks.

### H.3   Clean-label attack on MNIST $\rightarrow$ MNIST_M

For this experiment, 1% poison data is used to prevent the alignment of a target test point to its correct class. We test the attack on the binary classification problems (3 vs 8). Two approaches to initializing the poison data are evaluated. In the first approach, the poison data is initialized from the source domain data, and in the second approach, it is initialized from the target domain data. In both cases, the poison is picked from the class opposite to the true class of the target test point. Moreover, the poison data is initialized using the points closest in the input space to the target test point. The poison data obtained after solving Eq. 4 is added to the source domain data and UDA methods are retrained from scratch. The attack is considered successful if the target test point is misclassified after this retraining. For the results shown in Fig. 7, we randomly targeted 20 points and obtained poison data corresponding to each UDA method. Attack success is reported after evaluating UDA methods on five random initializations by adding the generated poison data in the source domain. To control the amount of maximum distortion between experiments, we add a constraint on the maximum permissible distortion to poison data using $\ell_\infty$ norm and use a value of $\epsilon = 0.1$. The poison data obtained after solving the optimization with base data chosen from the source and target domains with DANN as the UDA method are shown in Fig. 11. To generate poison data that remains effective even after UDA methods are trained from scratch, we make use of multiple randomly initialized networks during poison generation. Following the work [11], we reinitialize the models at different points during optimization. This re-initialization scheme helps train UDA methods with different random initializations and for a different number of epochs making the poison data more resilient to initialization change that can happen at test-time.

---

[3]https://bit.ly/34EFb52