# OpenReview forum: "Understanding the Limits of Unsupervised Domain Adaptation via Data Poisoning"
_NeurIPS.cc/2021/Conference — NeurIPS 2021 Poster_

### Official Review · Reviewer_Hs6Q · 2021-07-07

**Rating:** 6
**Confidence:** 4

**Summary:**

This paper aims to understand the limitation of unsupervised domain adaptation (UDA) through data poisoning. The paper reveals that the modern UDA deep learning approach is vulnerable to noise data, with almost zero accuracies in the poisoning target samples.

-----------Updates after Discussion

I appreciate the efforts and additional experimental results by the authors. Thus I recommend a (weakly) positive score.

**Limitations And Societal Impact:**

- The authors have adequately addressed the limitations and potential negative societal impact.

**Main Review:**

### Pros:
- This paper reveals an important and interesting fact that the modern UDA approach is quite vulnerable in the noisy datasets (particularly the poisoned target samples), which restricts its real practical utility.

### Cons:
- Despite the empirical discovery being interesting and promising, the current version seems preliminary and the in-depth analysis is lacking.
- The proposed theoretical result is insufficient to understand the inherent limitation.

Overall it is a good paper in the proof of concept but the main discovery is not surprising. Moreover, the in-depth analysis/justification/proof is not sufficient. Thus, I recommend a weak rejection but encourage a major revision for future submission.


### Comments:
This paper has several discoverings, the following are the comments for each contribution.

- “a lower bound on the target domain error providing a necessary condition for the success of learning in a UDA setting. The bound shows the failure of learning a domain invariant representation that minimizes source domain error at guaranteeing target generalization”.

In fact, this discovery is not surprising and several recent works have a good discussion on this idea. e.g ref [1,2], which is missing from the author. Besides, the theoretical proof is only valid for the bijective or invertible representation function (see Line 83 the definition). In fact, the most present deep UDA approach ** is not satisfied** this assumption. That being said, theory and practice have several non-ignorable gaps.

As for the conclusion of Theorem 1, the results do not provide novel insights, the lower bound can be vacuous in the presence of noisy labels. Supposing we have a MLP network the noisy source error is $e_s(h)>0$, then the lower bound can be still zero. From this perspective, the lower bound has some issues.

Line 40, the author states that “a large upper bound does not guarantee failure of UDA methods.“ I agree with this point. However, the upper bound is introduced as **sufficient condition** to ensure it works. Since the UDA only works in very specific scenarios, in the practical algorithm we need to justify the sufficient condition rather than the necessary assumption.

Another important concern of the proposed theoretical analysis is that the data generation procedure should be stochastic rather than a deterministic labeling function $f$. This is particularly important in this paper. The later experimental settings are exactly the stochastic setting: the same digit input $x$ can have two different values with 0 and 1. In contrast, the deterministic setting has no random sampling procedure (i,e for the same X, f(x) is fixed), which clearly violates the empirical settings.

- “ present example distributions where UDA succeeds, fails, and where success and failure are equally likely. The last case opens door to adversarial attacks such as data poisoning.”

In fact, the first two counterexamples are essentially the same as the ones in Ref [1]. Thus they are not surprising. As for the last counterexamples, the problem lies in the inherent property in UDA and is impossible to solve without additional assumptions/conditions. The authors are strongly encouraged to discover these conditions and justify them.


- “several novel data poisoning attacks….The poisoning attacks can be used to obtain a better sense of the robustness of the future UDA methods beyond simple benchmark datasets”

In general, this part is quite interesting and deserves further attention in the community. And I have the following question:

If we added the target sample noise to the source, the UDA-based approach will tend to find a similar point to learn the target. Naturally, the performance will have a significant drop in the shallow network since it is almost the worst-case attack. My concern is in the worst case, no algorithm can probably work. Thus it is an impossible task?

As far as I understand, the target sample attack is believed to be the most interesting and promising direction. (other types of attacks are quite common and fairly easy to solve by adopting the existing approaches) Thus I do think an additional discussion about the assumption, the condition is important for the future version.

### Other Comments

- The writing and organization can be improved. Sec 3.2 has only one paragraph and is relatively difficult to read.
- The performance drop in the digits and large dataset (e,g office) have quite different behavior. An in-depth analysis is expected.
- Line 142-144, the overparameterized network does not necessarily guarantee an expected term as zero. A rigorous justification is expected.

Reference
[1] Support and Invertibility in Domain-Invariant Representations.

[2] Representation bayesian risk decompositions and multi-source domain adaptation



**Time Spent Reviewing:**

6

---

> ### Author Response · Authors · 2021-08-10
> **Response to reviewer Hs6Q’s questions**
>
> > 1. “a lower bound on the target domain error .. That being said, theory and practice have several non-ignorable gaps.
>     Another important concern of the proposed ... which clearly violates the empirical settings.
>
>
> We thank the reviewer for pointing out the missing references. We have provided a detailed comparison of our work with them in the joint response above and we will appropriately cite these in our work. We would like to emphasize that we have not claimed that our work is the first one to show the failure of UDA methods. Instead, our work is focused on analyzing the reasons behind this failure (which we demonstrate by providing a simple lower bound on the target error) and proposing a theoretically motivated practical method to evaluate the robustness of UDA methods via our novel data poisoning attacks.
>
> The setting of our work is exactly the same as was used by Ben-David et. al (2007). Note that \\(g^{-1}\\) in line 83 (and also in Ben-David et al) means a preimage and not an inverse. Invertibility of \\(g\\) is unnecessary for Ben-David’s upper bound nor for our lower bound. If \\(g\\) were invertible, then \\(f(z)\\) would simply be \\(f(g^{-1}(z))\\) instead of \\(E[f(x)|g(x)=z]\\).
>
> As was also pointed out by Ben-David et al., , the labeling function \\(f\\) could be deterministic but the induced labeling function \\( \tilde{f}(z)\\) as defined in our Eq. 1 can be stochastic due non-invertibility of g, and it does not pose any problem for analysis.   We are unsure about the reviewer’s comment on empirical settings and would appreciate further clarification.
>
>
>
> > 2. As for the conclusion of Theorem 1, the results do not provide novel insights, the lower bound can be vacuous in the presence of noisy labels. Supposing we have a MLP network the noisy source error is es(h)>0, then the lower bound can be still zero. From this perspective, the lower bound has some issues.
>
> We respectfully disagree with the reviewer that our lower bound does not provide novel insights. Compared to previous works (see related work and the joint response) our lower bound is assumption-free and non-vacuous in scenarios when marginal label distributions of the source and target domains are the same (unlike previous works). Our data poisoning attacks which show the provable failure of UDA methods are also motivated from the lower bound. Thus we believe that our lower bound is a crucial step in the direction of understanding the limitations of learning in the UDA setting.
>
> We would like to clarify the case of \\( e_S(h) \geq 0 \\) as pointed out by the reviewer. In this case, our lower bound could be vacuous but so will be the upper bound proposed by Ben-David et. al (2009). The upper bound of Ben-David et. al, being large implies that the success of UDA methods cannot be guaranteed. Moreover, the lower bound proposed by Zhao et. al (\\( e_T(h) \geq (d_{JS}(p_S^Y, p_T^Y) - d_{JS}(p_S^Z, p_T^Z ))^2 - e_S(h) \\)),  could also be vacuous even if the marginal label distributions differ between the source and target. Hence we believe that the insight provided by our lower bound on the importance of matching conditional label distributions in the representation space between the source and target domain is a significant step towards improving UDA methods.
>
> Finally, in scenarios where  \\( e_S(h) \geq 0 \\), even the single domain problem cannot be fully solved and solving the UDA problem will be even harder. Hence it would not be very practical to use UDA methods in such a situation.
>
>
>
> > 3. Line 40, the author states that “a large upper bound does not guarantee failure of UDA methods.“ I agree with this point. However, the upper bound is introduced as **sufficient condition** to ensure it works. Since the UDA only works in very specific scenarios, in the practical algorithm we need to justify the sufficient condition rather than the necessary assumption.
>
> We agree with the reviewer that the success of UDA methods can be explained with the upper bound. However, since our work studies the failure of UDA, a necessary condition such as our lower bound is needed to explain the failure.
>
>
>
> > 4. In fact, the first two counterexamples are essentially the same as the ones in Ref [1]. Thus they are not surprising. As for the last counterexamples, the problem lies in the inherent property in UDA and is impossible to solve without additional assumptions/conditions. The authors are strongly encouraged to discover these conditions and justify them.
>
> We provided a detailed comparison of the purpose and rigor we used to study the cases presented in Fig. 1 in the joint response above. We hope that it clarifies the differences between the examples shown in Johansson et. al and our work. Case 3 where success and failure of UDA are equally likely has been thoroughly explained in section 3.2 in light of the lower bound. Furthermore, we have presented novel data poisoning attacks to precisely study the scenarios where UDA methods will provably lead to failure. In section 4 of the paper, we discuss in detail how poisoning leads various state-of-the-art UDA methods to learn a representation that leads to a large error on the target domain. Thus, we believe our theoretically motivated data poisoning attacks shed light on conditions/scenarios that lead to the failure of UDA methods.
>
>
>
> > 5. If we added the target sample noise to the source, the UDA-based approach will tend to find a similar point to learn the target. Naturally, the performance will have a significant drop in the shallow network since it is almost the worst-case attack. My concern is in the worst case, no algorithm can probably work. Thus it is an impossible task?
>
> Our data poisoning attacks show that UDA methods are extremely vulnerable to poisoning attacks. However, the varying initial divergence between the source and target domains as well as the percentage of poisoned data used, lead to the differences in the performance of various methods. Hence, defending against the attack is not impossible; however, our attacks do pose a significant challenge for the current UDA methods. It is for this reason that we believe researchers can use our poisoning attacks to evaluate the performance of future UDA methods on data distributions not conducive to UDA. Thereby helping them gain better insights into the performance of their methods than can be obtained from the evaluation on standard benchmark datasets.
>
>
>
> > 6. As far as I understand, the target sample attack is believed to be the most interesting and promising direction. (other types of attacks are quite common and fairly easy to solve by adopting the existing approaches) Thus I do think an additional discussion about the assumption, the condition is important for the future version.
>
> We are encouraged that the reviewer finds our poisoning attacks very interesting, especially the attack where we use mislabeled target domain data as poison points. We hope that our response to your comment 1 above would have clarified the misunderstanding of the invertibility assumption.
>
>
>
> > 7. The performance drop in the digits and large dataset (e,g office) have quite different behavior. An in-depth analysis is expected.
>
> This has already been justified in section 4.1 of the main paper. The main reason for the difference in the behavior of poisoning is attributed to the fact that for the Office dataset a representation pretrained on a clean imagenet dataset is used and the representation is only fine-tuned using UDA methods. On the other hand, for the Digits dataset, the representations are trained from scratch. Due to this difference, the performance of poisoning is inferior to that on the Digits dataset. Nevertheless, poisoning still leads to a significant reduction in the target domain accuracy on the Office dataset as shown in Table 2.
>
>
>
> > 8. Line 142-144, the overparameterized network does not necessarily guarantee an expected term as zero. A rigorous justification is expected.
>
> By this, we meant a small value and not necessarily zero exactly. The purpose of the line was to convey that with overparameterized networks and sufficiently large datasets UDA methods that aim to minimize the empirical estimates of the quantities in Eq. 3 can minimize the expected quantities. We will clarify the line in the updated version.

---

### Official Review · Reviewer_ErCq · 2021-07-15

**Rating:** 6
**Confidence:** 5

**Summary:**

In this work, the authors aim to study the limitations of Unsupervised Domain Adaptation through data poisoning. Specifically, the authors first show the failure of UDA by proving a simple lower bound, then present example distributions where UDA succeeds or fails. Finally, they propose several novel data poisoning attacks that use clean-label and mislabeled points to fail existing UDA methods and verify these attack methods in the experiments on two benchmark datasets.

**Limitations And Societal Impact:**

The authors claim that they have addressed the limitations of this work in Section 3 but I cannot find it. Indeed, they introduce two potential future directions, like extending the simple lower bound to different divergence metrics and boosting the poisoning attack by solving the bilevel problem.

**Main Review:**

Pros:
As far as I’m aware, this work is the first to explore the limitations of UDA methods from the lens of robustness. I believe this work could provide new insight for the community of UDA.
The code is available with the submission which improves the reproducibility index of the paper.
Overall, this paper is well-written and easy-to-follow.

Cons:

The relationship between data poisoning and the natural failure of UDA methods is not clear. From the view of robustness, supervised learning methods are still vulnerable under adversarial attacks like data poisoning, even though there are enough labeled examples from the target domain. It is not surprising that UDA methods are also vulnerable to data poisoning. Therefore, I am not sure that the fact that UDA methods are easy to be attacked can demonstrate the natural failure problem of UDA methods. If the authors could provide a convincing explanation during rebuttal, I will consider improving my score on this work.


---

The authors have addressed my concerns during rebuttal, however, the writing in the manuscript is still problematic. Thus I decide to improve my score to 6 here.

**Time Spent Reviewing:**

6

---

> ### Author Response · Authors · 2021-08-10
> **Response to reviewer ErCq’s questions**
>
> >1. The relationship between data poisoning and the natural failure of UDA methods is not clear. From the view of robustness, supervised learning methods are still vulnerable under adversarial attacks like data poisoning, even though there are enough labeled examples from the target domain. It is not surprising that UDA methods are also vulnerable to data poisoning. Therefore, I am not sure that the fact that UDA methods are easy to be attacked can demonstrate the natural failure problem of UDA methods.
> If the authors could provide a convincing explanation during rebuttal, I will consider improving my score on this work.
>
>
> Previous works on data poisoning have considered the fully supervised setting where train and test sets are drawn from the same underlying data distribution (single domain setting). Most works [10, 22, 26] have shown the difficulty of poisoning classifiers such as deep neural networks in this setting. These works either target the classification of a single test point by adding a large number of poisoned data or require modifying all points in a class to affect the model's performance on that class after retraining. We on the other hand show that poisoning in the UDA setting is significantly easier than in the single domain setting. As pointed out in section 3.2 where minimizing Eq. 3 can lead to an increase or decrease in \\(max \ (e_S( \tilde{f}_S, \tilde{f}_T ), \ e_T ( \tilde{f}_S, \tilde{f}_T ))\\) term of the lower bound. We have demonstrated poisoning as a concrete method to make UDA methods learn a representation with high target domain error. The significance of poisoning has been clarified in the joint response above along with how it provably leads to failure of UDA. We believe the fact that without poisoning UDA methods achieve high target domain accuracy whereas in presence of just a small percentage of poison data the target accuracy drops below 10\% to be extremely surprising. The failure implies that learning in the UDA setting is significantly more challenging than is projected by current UDA methods achieving high performance on benchmark datasets. Our poisoning attacks open up the possibility of exploring other ways to make UDA methods fail but more importantly provides a practical way for researchers to evaluate the robustness of their UDA algorithms against data distributions not conducive to learning in the UDA setting.

---

### Official Review · Reviewer_vTUY · 2021-07-16

**Rating:** 6
**Confidence:** 3

**Summary:**

The paper addresses the vulnerability of domain adaptation to adversarial attacks. First, a lower bound on the target risk of a given classifier is derived, which provides necessary conditions for the success of domain adaptation methods. Then, 3 examples on 2D data are given and analyzed to show the potential failure of DA methods aiming at aligning both domains while remaining good for classification on the source one. The last example describes an ambiguous situation for which success and failure of adaptation are equally likely, from which the authors have the inspiration that adding a few poison data can easily favor the bad solution. They concretize this idea by proposing 3 data poisoning strategies and they show their ability to deteriorate the performance of previously proposed DA methods.

**Ethical Concerns:**

.

**Limitations And Societal Impact:**

Although the societal impact was not directly addressed in the paper, the aspects of the adversarial attack it tackles are vital in real-world applications. For example, in autonomous driving, it can be very problematic for an algorithm to adapt to new environments if it is not robust to data poisoning.

**Main Review:**

The paper is clearly written and easy to follow, and the theoretical technical results are supported by proofs.

## Strengths
* The overall idea of showing the vulnerability of DA methods to adversarial attacks is interesting. It illustrates the intuition that the complete absence of target data is a considerable lack of information for DA approaches methods. Also, the provided empirical evidence for this vulnerability concerns even new DA approaches that aim to improve previous ones (IW-DAN and IW-CDAN).
* Corollary 3 is very insightful as it offers more precision on which quantity is being minimized by DA methods that try to have a good performance on the source domain while embedding the source and target domain into a representation space in which they are aligned.

## Weaknesses
* About lower bounds on the ideal joint error: in line 39, it is argued that a large upper bound does not guarantee the failure of DA methods. However, any lower bound on the ideal joint hypothesis is equivalent to a lower bound on the target risk, due to the equivalence between
$\min_{h \in H}e_S(h) + e_T(h) \geq B$ and $\forall h \in H, e_T(h) \geq B - e_S(h)$, where $B$ denotes some lower bound.
* Lack of comparison to previous work on the drop of performance on the target domain due to the deep embedding of both domains
  * Comparison to the lower bound established in Theorem 4.3 of  [1]: using the notation of the paper being reviewed, their result states that for any $h \in H, g \in G$, $e_T(h) + e_S(h)\geq\frac{1}{2}(d_{JS}(\tilde p_S, \tilde p_T)-d_{JS}(p_S^Y, p_T^Y))^2$ (where $p_S^Y$ and $p_T^Y$ denote the label marginal distributions), whereas Theorem 1 of the reviewed paper states that $e_{T}(h) + e_{S}(h) \geq \max (e_S(\tilde{f}_S, \tilde{f}_T), e_T(\tilde{f}_S, \tilde{f}_T)) - D_1(\tilde{p}_S, \tilde{p}_T)$. The r.h.s. of the former inequality is always nonnegative, which is not the case for the latter. This comparison should be carried out in detail.
  * Another contribution concerned with the performance drop on the target domain after the embedding in the new representation space is [2]. In particular, their Example 1 illustrates a case with two global solutions to the classic DA minimization problem, which is the same spirit of Case 3 of Figure 1 of the paper under review.


## Suggestions
* Detailed comparison to [1,2].
* Plotting  2D map of the effect of both the percentage of poisonous data and the divergence between domains.

## Comments
* In line 232,  the "generator" term should be replaced with "source classifier" for example. Indeed, in spite of the similarity in mechanism between GANs and adversarial domain adaptation methods, they are not the same. This holds also for line 242
* Having seen Figure 1, in which the data is represented in a 2D space, I was confused when looking at Figure 3 for a moment. Indeed, it seems like the target classes were flipped from left to right, whereas the arrow represents an alignment in a new representation. In this case, another schematic representation for the data in the new representation space should be used.
* I do not see the interest of Corollary 2 concerning the KL-divergence, as it states a looser lower bound than the one of Theorem 1 (as $-D_1(\cdot,\cdot) \geq -\sqrt{\frac{1}{2}D_{KL}(\cdot\Vert \cdot)}$), and it does not have other interests shown in the paper, besides involving another divergence.
### Minor comments

* In lines 168 and 170, it should be $e(\tilde f_S, \tilde f_T)$ instead of $e(\tilde f_S, \tilde f_S)$.
* Line 252: "poisoned data, leads" --> "poisoned data leads"
* Line 253: ". There by leading " --> ", thereby leading"
* Line 288: "Fig. 9 ... Fig. 8" ---> "Fig. 10 ... Fig. 9"
* Line 338: "showe" --> "show"
* Line 340: "presence of even small" --> "the presence of even a small"

## Questions
* In the third case of examples in section 3.2, the classifier is initialized randomly, and this randomness causes the optimization to converge to either a good or bad solution. Do you have an idea about the initial conditions of the solution that lead to one solution or the other? (e.g. a partition of two regions in the solution space according to which the optimization results in one of the two minimizers).

#  References

[1] Zhao, H., Combes, R. T. D., Zhang, K., & Gordon, G. (2019). On Learning Invariant Representations for Domain Adaptation. *International Conference on Machine Learning*, 7523–7532.

[2] Johansson, F. D., Sontag, D., & Ranganath, R. (2019). Support and Invertibility in Domain-Invariant Representations. *The 22nd International Conference on Artificial Intelligence and Statistics*, 527–536.

**Time Spent Reviewing:**

4

---

> ### Author Response · Authors · 2021-08-10
> **Response to reviewer vTUY’s questions**
>
> >1. “About lower bounds on the ideal joint error: in line 39, ... equivalent to a lower bound on the target risk, due to the equivalence between \\( min_{h \in H} e_S(h) + e_T(h) \geq B \)) and ∀h∈H,eT(h)≥B−eS(h), where B denotes some lower bound”.
>
> We agree with the reviewer that if the error of the ideal joint hypothesis increases then the target risk of any hypothesis increases in the UDA setting. However, the point we wanted to make in line 39 was about the upper bound on the target risk proposed by Ben-David et. al, which can only be used to explain the success of UDA methods but not their failure (as agreed by reviewer Hs6Q too). We understand the confusion the current wording could lead to and we will clarify it in the paper.
>
> The lower bound suggested by the reviewer using \\( min_g,h  e_s(h)+e_t(h) \\) is mathematically (and trivially) correct.  It is the minimum over any classifier \\(h\\) and representation \\(g\\). Thus, when its value is large the problem is inherently difficult and any UDA is futile. However, when the two domains are only moderately different, this minimum value can be very small especially with a big hypothesis space like deep neural networks. In contrast, our bound can still suggest the UDA failure in this case, because the \\( h \\) and \\(g\\)that minimize Eq. 3 using a UDA method can be very different from the \\( h \\) and \\(g\\) that minimize the ideal joint error. Thus giving us a large value \\(e[|\tilde{f}_S - \tilde{f}_t|]\\) in our bound. Moreover, we would also like to emphasize that our lower bound is not obtained using the equivalence pointed by the reviewer (see Appendix A for the proof) even though the final lower bound has a similar form of   \\( e_T(h) \geq B - e_S(h) \\). Thus, our bound cannot be trivially obtained from existing results and is more informative in the UDA setting than the bound suggested by the reviewer.
>
> >2. Comparison to the lower bound established in Theorem 4.3 of [1]: ... The r.h.s. of the former inequality is always nonnegative, which is not the case for the latter. This comparison should be carried out in detail.
>
> We agree with the reviewer that in cases where domain adaptation is not done i.e. Eq. 3 is not minimized, our lower bound could be negative and hence vacuous. However, the lower bound is specifically useful for scenarios where UDA methods are used to minimize Eq. 3 (the focus of this paper) which leads to making \\( e_S(h) = 0 \\) and \\( D_1(\tilde{p_S}, \tilde{p_T}) = 0 \\).  In this case, our lower bound will give a non-negative result. We would like to point out that unlike our lower bound which is assumption-free, the bound presented in Theorem 4.3 of [32] is informative only when \\( d_{JS}(p_S^Y, p_T^Y) > d_{JS}(p_S^Z, p_T^Z) \\) i.e. there is a difference in the marginal label distribution between the source and target domains and domain adaptation is being done to minimize \\( d_{JS}(p_S^Z, p_T^Z) \\) where \\(Z\\) is the representation. As explained in detail in our joint response above, the bound of [32] is applicable in specific situations (\\(d_{JS}(p_S^Y, p_T^Y) > d_{JS}(p_S^Z, p_T^Z)\\)) only whereas our lower bound gives better estimate of the target risk when \\(d_{JS}(p_S^Y, p_T^Y)=0\\).
>
> >3. Another contribution concerned with the performance drop on the target domain after the embedding in the new representation space is [2]. In particular, Example 1 illustrates a case with two global solutions to the classic DA minimization problem, which is the same spirit of Case 3 of Figure 1 of the paper under review.
>
> We thank the reviewer for pointing out the missing reference and we will appropriately cite it. We provide a detailed comparison of the purpose and the rigor with which we dealt with the example in our paper and its use to validate our lower bound as well as motivate our data poisoning attacks in the joint response above. We hope that it clarifies the differences between our illustrative cases and example 1 in [Johansson2019].
>
>
>
> >4. Detailed comparison to:
>     [32] Zhao, H., Combes, R. T. D., Zhang, K., & Gordon, G. (2019). On Learning Invariant Representations for Domain Adaptation. _International Conference on Machine Learning_, 7523–7532.
>     [Johansson2019] Johansson, F. D., Sontag, D., & Ranganath, R. (2019). Support and Invertibility in Domain-Invariant Representations. _The 22nd International Conference on Artificial Intelligence and Statistics_, 527–536.
>
> We have provided a detailed comparison of our work with the two papers in the joint response above. We hope that it clarifies how our lower bound provides non-vacuous results beyond the setting of different marginal label distributions between source and target domains considered by these works. Furthermore, the ease with which data poisoning can help bring out the limitations of UDA without requiring strict assumptions on the data distributions of the source and target domains is another strength of our work that distinguishes it from these works.
>
>
>
> >5. Plotting 2D map of the effect of both the percentage of poisonous data and the divergence between domains.
>
> We use a simple two moons dataset to illustrate the effect of changing the poison percentage and changing the divergence between the two domains. The target domain data is the translated version of the source domain data along the x-axis. The amount of translation is used to measure the divergence between the source and the target domains. For this experiment, we use mislabeled target domain data as our poison points. This is referred to as wrong-label incorrect-domain poisoning in the main paper. Consistent with the results present in rows marked with Poison_target of Table 1 and 2 of the main paper and Table 4 in the Appendix, we see that increasing the divergence between the source and target domains makes it easy to poison UDA methods with small amounts of poisoned data. Moreover, at a fixed divergence level increases the amount of poisoned data leads to a larger drop in the target accuracy.
>
> | Method | Dataset | Target | domain | accuracy|
> |--------|---------|------------------------|----------------|-----------------|
> |  |  | Divergence=0.25 | Divergence=0.5 | Divergence=0.75 |
> Source Only|Clean |99.6$\pm$0.0|87.92$\pm$0.96|68.40$\pm$0.13|
> ||Poisoned (5%)|79.76$\pm$2.94|55.24$\pm$4.42|59.44$\pm$2.77|
> ||Poisoned (10%)|53.16$\pm$2.44|46.52$\pm$3.51|42.56$\pm$2.65|
> |  |  |  |  |  |
> DANN[9]|Clean |98.48$\pm$1.45|97.24$\pm$1.27|71.80$\pm$3.92|
> ||Poisoned (5%)|95.36$\pm$2.54|83.08$\pm$1.43|62.72$\pm$3.63|
> ||Poisoned (10%)|85.52$\pm$10.01|65.56$\pm$8.44|40.92$\pm$5.97|
> |  |  |  |  |  |
> CDAN[16]|Clean |100.0$\pm$0.0|94.52$\pm$1.30|76.08$\pm$4.49|
> ||Poisoned (5%)|91.36$\pm$2.98|77.48$\pm$6.23|64.28$\pm$4.95|
> ||Poisoned (10%)|87.12$\pm$1.35|73.16$\pm$0.8|46.04$\pm$8.73|
> |  |  |  |  |  |
> MCD[25]|Clean |100.0$\pm$0.0|91.88$\pm$2.35|69.28$\pm$2.77|
> ||Poisoned (5%)|79.68$\pm$5.52|70.48$\pm$10.69|58.88$\pm$9.96|
> ||Poisoned (10%)|65.8$\pm$2.66|44.28$\pm$5.73|40.96$\pm$16.42|
> |  |  |  |  |  |
>
> > 6. Typos:
>
> We thank the reviewer for pointing these out and we will fix them in the next version.
>
>
>
> > 7. In the third case of examples in section 3.2, the classifier is initialized randomly, and this randomness causes the optimization to converge to either a good or bad solution. Do you have an idea about the initial conditions of the solution that lead to one solution or the other? (e.g. a partition of two regions in the solution space according to which the optimization results in one of the two minimizers).
>
> As discussed in section 3.2, case 3 is an ambiguous case where there are two optimal solutions to Eq. 3. Due to the presence of two solutions, local methods such as SGD may converge to any of them. We believe the failure of UDA methods is more fundamental and cannot be fixed by changes in the initialization. To carefully analyze the precise question of when UDA fails we use our data poisoning attacks. As explained in section 4 with the addition of mislabeled target domain data in the source the discriminators of DANN and CDAN are optimal as long as source and target domain data are aligned (irrespective of the labels). However, minimization of the source domain classification loss (\\(e_S\\)) leads discriminator-based UDA to align incorrect classes from the two domains together. Thus leading to high target domain error. The effect of poisoning on other UDA methods is discussed in detail in section 4 of the main paper.

---

### Author Response · Authors · 2021-08-10
**Joint response to all the reviewers:**

**Importance of our lower bound and its comparison to previous works:**

* **Our lower bound does not make any assumptions on the data distributions:** Unlike the previous work [32], which presented an information-theoretic lower bound to explain the failure of learning a domain invariant representation when marginal label distributions differ across the source and target domains, our lower bound does not require any such assumption. Thus, the bound presents a necessary condition for successful learning in the UDA setting. In particular, our lower bound shows that a UDA method may succeed or fail at target generalization i.e. the term \\(max \ (e_S( \tilde{f}_S, \tilde{f}_T ), \ e_T ( \tilde{f}_S, \tilde{f}_T ))\\) in our lower bound (Eq. 2) can be small or large, even when a domain invariant representation is learnt that minimizes source error.

* **Our lower bound provides better insights into the failure of UDA than existing bounds:** Unlike the bounds presented by previous works [32, Johansson2019] which fail to provide insights into the behavior of UDA when marginal label distributions are the same across the two domains, our bound remains tight.  In particular, the information-theoretic lower bound of [32] (Theorem 4.3 of [32]) becomes vacuous when \\( d_{JS}(p^Y_S, p^Y_T) = 0 \\). However, our lower bound will be non-vacuous in this scenario as long as UDA methods are used i.e., \\( e_S \\) and \\( D_1(\tilde{p}_S, \tilde{p}_T)\\) are minimized. A concrete example where our bound is tighter than the bound of Theorem 4.3 of [32] is their example in section 4.1 where the true target risk is 1. As discussed by the authors themselves (last paragraph below Corollary 4.1), their lower bound on the example is vacuous and says that the target risk will be greater than 0. In comparison, our lower bound in Theorem 1 says that the target risk is greater or equal to 1 which is tighter and much more informative. Moreover, our lower bound is also more informative than the bound suggested by reviewer vTUY which although mathematically correct does not provide much intuition into the behavior of UDA methods since it assumes the representation and the classifier are learned using labeled target domain data which violates the conditions of UDA.

* **Explanation of failure of UDA methods is consistent with previous works:** Compared to the previous work of [Wu2020] which decomposed the target risk into source risk, representation conditional label divergence, and representation covariate shift and explained the reason for the failure of DANN to be its inability to account for representation conditional label divergence. Our lower bound suggests a similar explanation for the failure of DANN i.e, without access to any labeled target domain data, DANN may learn a representation that induces labeling functions (\\(\tilde{f}_S\\)) and target (\\(\tilde{f}_T\\)) which do not agree with each other on the source and target domains. This difference in the induced labeling functions is captured by \\(max \ (e_S( \tilde{f}_S, \tilde{f}_T ), \ e_T ( \tilde{f}_S, \tilde{f}_T ))\\) term in our lower bound. Our explanation for UDA failure extends beyond DANN as we illustrated through our data poisoning attacks, which are concrete ways to lead UDA algorithms to learn a representation that incurs high error on the target domain.

**Significance of our illustrative cases and their comparison to Example 1 of [Johansson2019]:**

* **Illustrative cases are used for motivation but our main contribution is the lower bound and data poisoning attacks:** We would like to clarify that the goal of this work was to propose a necessary condition for successful learning in the UDA setting and to demonstrate the vulnerability of UDA to data poisoning. We used the illustrative cases to demonstrate the existence of three possible scenarios that may be encountered when learning in the UDA setting (by minimizing Eq. 3) based on the divergence between the source and target domains. Case 3, where success and failure of UDA are equally likely, illustrates that without any labeled target domain data success or failure of UDA cannot be predicted (i.e., the behavior of \\(max \ (e_S( \tilde{f}_S, \tilde{f}_T ), \ e_T ( \tilde{f}_S, \tilde{f}_T ))\\)term in the lower bound is unknown) and small misinformation about target domain labels can lead UDA methods to the case where they learn a representation that fails to generalize on the target domain. Thus, this case is used as a motivation for our novel data poisoning attacks.

* **We provide an in-depth treatment of the illustrative cases using both analytical and numerical methods:** Unlike [Johansson2019] which used their Example 1 to point out the existence of two solutions to Eq. 3 in a 2D case. We have explicitly computed the representation that will be learned by UDA methods (full details are shown in Appendix B) by assuming source and target domain distributions as Gaussian mixtures rather than using projection onto the x or the y axis as our representations. We go on to numerically evaluate each of the terms that appear in our lower bound in Theorem 1 and show the target risk values obtained from the bound in each of the three cases. Thus, we believe our rigorous and in-depth analysis of the illustrative cases helps explain the validity of our lower bound as well as motivate our novel data poisoning attacks.

**Significance of our data poisoning attacks against UDA methods:**

* **Poisoning in the UDA setting is surprisingly easier than in the single domain setting:** Most works [10, 22, 26] have shown the difficulty of poisoning classifiers such as deep neural networks in the fully supervised setting where train and test sets are drawn from the same underlying data distribution (single domain setting). These works either target the classification of a single test point by adding a large number of poisoned data or require modifying all points in a class to affect the model's performance on that class after retraining. On the other hand, we find that poisoning in the UDA setting is much easier than in the single domain setting. Our poisoning attacks with mislabeled target domain data as poisons can reduce the target domain accuracy from greater than 60% to less than 10% on the Digits dataset with the addition of small amounts of poisoned data (6%-10%). Our watermarking and clean label attacks further improve the practicality of poisoning attacks against UDA and pave the way for future research in this direction, such as using data reordering or initialization-based attacks against UDA methods.

* **Poisoning leads to provable failure of learning with UDA methods:** Our data poisoning attacks present a practical way of fooling UDA methods into learning a representation that incurs high error on the target domain. The representation learned by UDA methods in presence of poisoned data makes the  \\(max \ (e_S( \tilde{f}_S, \tilde{f}_T ), \ e_T ( \tilde{f}_S, \tilde{f}_T ))\\) term large in our lower bound suggesting that poisoning leads to a provable failure of UDA methods.

* **Poisoning highlights the flaws with current benchmark datasets used for evaluating UDA methods:** The apparent failure of UDA methods in presence of a small percentage of poisoned data makes the usefulness of the existing UDA methods and evaluation benchmarks questionable especially in the scenarios where source/target data distributions are not conducive for UDA (case 3 of our Fig. 1). In section 4 of our paper, we describe how poisoning causes different UDA methods to fail (see our t-SNE plots and Figs. 3, 5, and 6). These novel poisoning attacks distinguish our paper from previous papers exploring the failure modes of UDA as well as work on data poisoning attacks. Firstly, our work provides a practical evaluation procedure for UDA methods that, as agreed by all reviewers, provides insights that can improve the robustness of UDA algorithms to distributions that are not favorable for learning in the UDA setting. Secondly, it goes beyond the commonly studied single domain setting in data poisoning and demonstrates that the difficulty of poisoning in this setting does not undermine the threat posed by poisoning to machine learning algorithms.

**New References:**

[Johansson2019] Johansson, F. D., Sontag, D., & Ranganath, R. (2019). Support and Invertibility in Domain-Invariant Representations. _The 22nd International Conference on Artificial Intelligence and Statistics_, 527–536.

[Wu2020] Wu, Xi, et al. "Representation Bayesian Risk Decompositions and Multi-Source Domain Adaptation." _arXiv preprint arXiv:2004.10390_ (2020).

---

### Author Response · Authors · 2021-08-10
**Summary**

We thank all the reviewers for their valuable feedback and insightful questions. We are encouraged that the reviewers found the main idea of our work on analyzing the limitations of unsupervised domain adaptation (UDA) methods from the lens of data poisoning to be important (Hs6Q), interesting (vTUY, Hs6Q), and capable of providing novel insights to improve future UDA methods (vTUY, ErCq, Hs6Q). The main concern of reviewers vTUY and Hs6Q is about the comparison of our work to previous works which explained the failure modes of UDA. Due to lack of space, we briefly compared our work with these works in lines 93-100 of the main paper. However, we have expanded on the related work of the paper and have highlighted how our work differs from these in the response below. On the other hand, the reviewer, ErCq, sought clarification on how data poisoning demonstrates the limitations of UDA methods. Although we discussed this question at length in section 4 of the paper, we provide further clarifications to address this concern below.

​​For answers to the specific questions raised by each reviewer please see our individual comments below. Based on these answers and clarifications we believe we have addressed the concerns of each of the reviewers. We are hopeful that reviewers would consider our answers and would increase their ratings and recommend acceptance. We are just a post away and will be happy to answer any follow-up questions the reviewers and area chair may have during the discussion phase.

---

### Comment · Area_Chair_xzZ6 · 2021-08-13
**Further questions for the authors**

Dear Reviewers,

If you have further questions, please post them here (not under the rebuttal because it will become so difficult to manage). The other thread (our committee discussion) is not visible to the authors. Please carefully distinguish these two threads.

Many thanks for your efforts!

Regards,
AC

---

> ### Comment · Reviewer_Hs6Q · 2021-08-15
> **Response: Rebuttal to Review Hs6Q**
>
> I have read the rebuttal and thanks for the detailed response. Some of my concerns are addressed. My additional feedbacks are as follows:
>
> > Poisoning in the UDA setting is surprisingly easier than in the single domain setting.
>
> Could you elaborate more details on this? Why does the performance on UDA with simple random labels surprisingly degrade? What is the key difference w.r.t. the conventional single task learning?
>
> > Poisoning highlights the flaws with current benchmark datasets used for evaluating UDA methods.
>
> I partially agree with this point. The empirical results indeed verify the failures on several popular benchmarks. Is this problem caused by the inherent limitation of distribution matching (i,e matching $P(x)$, $P(x|y)$ or $P(y|x)$ is incorrect) ? Or the proposed algorithms are not proper (i.e, matching P(x) is correct, but the algorithms such as DANN are problematic).

---

> > ### Author Response · Authors · 2021-08-17
> > **Response to Hs6Q's additional questions**
> >
> > We thank the reviewer for these insightful questions! Please see our responses to them below. We would be happy to provide any further clarifications if needed.
> >
> > >Poisoning in the UDA setting is surprisingly easier than in the single domain setting. Could you elaborate more details on this? Why does the performance on UDA with simple random labels surprisingly degrade? What is the key difference w.r.t. the conventional single task learning?
> >
> > The fully supervised single domain setting assumes that the training and test data are samples from the same distribution. In this setting, the presence of a small amount of label noise (1-10%) in the training data cannot significantly degrade the performance of classifiers such as deep neural networks due to the presence of a large amount of correctly labeled data. However, UDA methods rely only on the labeled source domain data and the matching of unlabeled marginal distributions of source and target domains. As explained in Case 3 of Fig. 1, a small number of target domain labels (true or poisoned) can have much sway over the results. Moreover, minimizing the loss on the poisoned source domain in this scenario leads to correctly labeling the poison data (which are mislabeled target domain data in wrong-label incorrect-domain poisoning), causing the classes in the target domain to be aligned with the wrong class of the source domain (as shown in the t-SNE plots in Fig. 2), which leads to a poor target domain generalization.
> >
> > Thus, relying on marginal distribution matching in absence of target domain labels (especially when the divergence between the source and target domains is large, see our answer to vTUY’s question 5) makes poisoning in the UDA setting fundamentally different (and easier) than in the single domain setting.
> >
> > >Poisoning highlights the flaws with current benchmark datasets used for evaluating UDA methods. I partially agree with this point. The empirical results indeed verify the failures on several popular benchmarks. Is this problem caused by the inherent limitation of distribution matching (i,e matching ,  or  is incorrect) ? Or the proposed algorithms are not proper (i.e, matching P(x) is correct, but the algorithms such as DANN are problematic).
> >
> > Our poisoning attacks demonstrate that evaluating the performance of UDA methods on current benchmark datasets does not provide us complete insights into their true performance as agreed by the reviewer Hs6Q. As demonstrated by our analysis (Theorem 1 and Sec. 3.2) as well as our empirical results (Sec. 4), UDA methods that rely on matching the marginal distributions to align the conditional label distributions of the source and target domains (e.g., DANN) are prone to failure. However, UDA methods that aim to minimize the mismatch between the conditional label distributions (e.g., CDAN, which uses pseudo-labels in the discriminator) can be more successful at learning in diverse UDA settings and would also be robust to poisoning attacks. For instance, results in Table 3 show that the performance drop of different UDA methods is significantly different (with CDAN being the most robust), even though all methods achieve impressive performance on the clean dataset. Thus, the robustness of a UDA method to poisoning attacks has the potential to translate into a better performance of the method on unknown distributions.
> >
> > Therefore, our theory and poisoning attacks highlight the shortcomings of current UDA methods, and our data poisoning attacks provide a simple evaluation procedure for future UDA methods to gauge their effectiveness on data distributions not conducive to UDA.

---

> > > ### Comment · Reviewer_Hs6Q · 2021-08-17
> > > **Follow-up**
> > >
> > > Thanks for your additional responses. For the second question, it is possible to provide additional empirical results (target poisoned on digits) on other kinds of distribution alignments? For example,
> > >
> > > - Learning Transferable Features with Deep Adaptation Networks. Long, ICML 2015
> > >  (This paper adopted MMD on the feature space without adversarial training.)
> > >
> > > - A DIRT-T Approach to Unsupervised Domain Adaptation. Shu, ICLR 2018
> > >
> > > (This paper adopted the robustness idea in unsupervised domain adaptation.)
> > >
> > > - CyCADA: Cycle-Consistent Adversarial Domain Adaptation. Hoffman, ICML 2018
> > >
> > > (This paper adopted the domain transformation rather than distribution matching.)
> > >
> > > I am really sorry for the additional experimental requirement. Because the original paper only tackles four baselines, thus additional experiments on another type of UDA training can significantly improve this paper.

---

> > > > ### Comment · Reviewer_ErCq · 2021-08-25
> > > > **Same concerns as Reviewer Hs6Q**
> > > >
> > > > My concerns are almost the same as those of Reviewer Hs6Q. From my point of view, it is not surprising to see that poisoning in the UDA setting is easier than standard training due to the weak information from the training dataset like weakly supervised learning. Therefore, I am also looking forward to seeing additional empirical results to strengthen the point that "Poisoning highlights the flaws with current benchmark datasets used for evaluating UDA methods."

---

> > > > > ### Author Response · Authors · 2021-08-25
> > > > > **Addressed concern in comment to Hs6Q**
> > > > >
> > > > > We thank you for your response. We present the new results in our comment to Hs6Q. We hope these new results clarify your concerns as to how poisoning provides better insights into the performance of UDA methods beyond the benchmark datasets. Please let us know if you have any further questions.

---

> > > > ### Author Response · Authors · 2021-08-25
> > > > **New Experiments**
> > > >
> > > > Below we present the results of using our poisoning attacks on UDA methods suggested by the reviewer Hs6Q. Our results of poisoning on these new experiments are consistent with the results reported in the original submission where we had shown that a small amount of poisoned data can lead UDA methods to learn a representation that fails to generalize on the target domain. Moreover, in cases where the source and target domains are very close (such that training on the source domain leads to high performance on the target without UDA) poisoning has limited effect. We agree with the reviewer that these additional results further improve the quality of our work and we will add these to the main paper. We hope the results of our experiments clarify the concerns of the reviewer as to how poisoning provides better insights into the performance of UDA methods beyond the benchmark datasets. Please let us know if you have any further questions.
> > > >
> > > > As per the Hs6Q’s suggestion, we used wrong-label incorrect-domain poisoning attacks where we add mislabeled target domain data as poisons in the source domain. The results presented below are an average of three different runs.
> > > >
> > > > > Learning Transferable Features with Deep Adaptation Networks. Long, ICML 2015 (This paper adopted MMD on the feature space without adversarial training.)
> > > >
> > > > Following the original and follow-up works on MMD [6], we also evaluated the effect of our poisoning attacks against this method using the Office-31 dataset. The results of poisoning on DAN with 10% mislabeled target domain data, consistent with the other results shown in Table 2 of the main paper, demonstrate our poisoning attack's effectiveness against this method. We used the Pytorch version of the implementation for DAN available from the official code of Combes et. al [6] to generate these results.
> > > >
> > > > |DAN|A->D|A->W|D->A|D->W|W->A|W->D|
> > > > | ----------- | ----------- | ----------- | ----------- | ----------- | ----------- | ----------- |
> > > > Clean|82.3|80.1|68.9|98.0|66.1|99.0|
> > > > Poisoned (10%)|59.8|51.4|6.7|53.7|7.6|79.9|
> > > > ||||||||
> > > >
> > > > > A DIRT-T Approach to Unsupervised Domain Adaptation. Shu, ICLR 2018 (This paper adopted the robustness idea in unsupervised domain adaptation.)
> > > >
> > > > Here we present the results of our poisoning attack against the DIRT-T method proposed by the paper, using all the Digits tasks we considered in our paper. This method used virtual adversarial training and conditional entropy minimization whose effectiveness is contingent on the quality of the pseudo-labels of the target domain data. In our work, we had already considered a method, CDAN, which relied on the idea of pseudo-labels for the target domain data. Consistent with our results on CDAN presented in Table 1, we see our poisoning attacks are effective at reducing the target domain accuracy for the DIRT-T method. We used 10% mislabeled target domain data as our poisons (same as that used in Table 1) and used a similar setting to the official implementation of the paper for our evaluation.
> > > >
> > > >
> > > > |DIRT-T|SVHN -> MNIST|MNIST->MNIST_M|MNIST->USPS|USPS->MNIST|
> > > > | ----------- | ----------- | ----------- | ----------- | ----------- |
> > > > Clean|92.45$\pm$0.46|91.41$\pm$0.16|98.15$\pm$0.28|98.13$\pm$0.09|
> > > > Poisoned (10%)|0.09$\pm$0.02|0.37$\pm$0.01|4.62$\pm$4.33|0.31$\pm$0.24|
> > > > ||||||
> > > >
> > > > > CyCADA: Cycle-Consistent Adversarial Domain Adaptation. Hoffman, ICML 2018. (This paper adopted the domain transformation rather than distribution matching.)
> > > >
> > > > This work uses a Cycle GAN coupled with a semantic loss dependent on the pseudo-labels for the target domain data, to enforce consistency of the two domains in the pixel space as well as in the representation space. When using the poisoned data to train the cycle GAN, we found that it failed to generate good transformations of the source domain. Due to limited time, we could not conclude whether the failure of cycle GAN was due to poisoning or due to hyperparameters.  Thus, in our poisoning experiment, we omitted training the cycle GAN and used the transformed version of the source domain images included in the official repository and treated them as our original source domain images. We present the results for poisoning only the feature level domain adaptation below (assuming cycle GAN is trained on clean data). The high target accuracy of training using only the transformed source domain data (transformed source only) compared to the target accuracy obtained using feature-level domain adaptation on clean data implies a high similarity between the transformed source and the target domains. Due to this high similarity between the transformed source and the target domains, our poisoning attacks (using 10% mislabeled target domain data as poisoned data) have limited effect. For USPS->MNIST, poisoning with 10% of MNIST training data (6000 images) is comparable to the size of USPS training data (7291 images) and thus we see a significant reduction in the target domain accuracy. These results are consistent with the results presented in the main paper, especially on tasks W->D and D->W in Table 2, where poisoning fails since the source and target domains are very similar (indicated by high source only performance).
> > > >
> > > > |CyCADA|SVHN -> MNIST|MNIST->USPS|USPS->MNIST|
> > > > | ----------- | ----------- | ----------- | ----------- |
> > > > Transformed Source Only|74.5$\pm$0.3|95.6$\pm$0.2|96.4$\pm$0.1|
> > > > Feature level (Clean data)|90.4$\pm$0.4|95.6$\pm$0.2|96.5$\pm$0.1|
> > > > Feature level (Poisoned data, 10%)|84.3$\pm$2.3|93.3$\pm$0.5|60.7$\pm$3.5|
> > > > |||||

---

> > > > > ### Comment · Reviewer_Hs6Q · 2021-08-25
> > > > > **Re: New Experiments**
> > > > >
> > > > > Dear authors,
> > > > >
> > > > > I appreciate the efforts for additional experimental results. Based on these, I increased my score to 6. Besides, I have the following additional suggestions,
> > > > >
> > > > > - The current empirical discovery seems much stronger and interesting. I hope the author can include them in the final draft.
> > > > > - I think major revision on the paper format and presentation (e.g, figures is better to use vector image rather than PNG) are necessary.
> > > > > - The theoretical discussion can be revised. The core empirical finding is sufficient to get acceptance. The additional theoretical analysis is not always necessary if it could not reveal the true reason. I would recommend the author to read the DomainBed paper, which is purely empirical but also quite interesting.
> > > > > https://openreview.net/forum?id=lQdXeXDoWtI
> > > > >
> > > > > The additional experiments made me believe in the strong merits and potential of this paper. I hope the author can improve the paper presentation by following the line of DomainBed, whether it gets accepted or not.

---

> > > > > > ### Author Response · Authors · 2021-08-25
> > > > > > **Updated score**
> > > > > >
> > > > > > We thank the reviewer for updating their rating of our work. We are encouraged that you found our data poisoning attacks to be interesting, of significant importance, and sufficient to get acceptance. We will use the answers to the raised questions, new experiments as well as your other suggestions to further improve the quality of our work.

---

> ### Comment · Reviewer_vTUY · 2021-08-24
> **Response to the authors**
>
> Dear authors,
>
> I have read the rebuttal and I thank you for having addressed most of the points I raised in detail.
> However, I would like to clarify that the first point I raised was just to imply that a lower bound on the joint error does indeed suffice to have a lower bound on the risk on the target domain. Also, I think the whole analysis with the $B$ term I introduced can be done after changing the data representation and considering $h$ as a classifier over the fixed representation $g$ (that can be learned by a neural network). In any case, the point I raised was not about the novelty of the bound presented in the paper, but just about the statement made in line 39.

---

> > ### Author Response · Authors · 2021-08-25
> > **Response to vTUY**
> >
> > We thank you for your response. We are glad that we were able to address your concerns and are hopeful that you will increase your rating of our work and champion our paper for acceptance. We will include your suggestions in the final version along with making the sentence in Line 39 clearer. Please let us know if you have any further questions.

---

### Decision · Program_Chairs · 2021-09-27

**Decision:**

Accept (Poster)

**Comment:**

This paper studies the vulnerability of domain adaptation to adversarial attacks. The key idea is to show the failure of UDA by proving a simple lower bound, then present example distributions where UDA succeeds or fails. This paper proposes several novel data poisoning attacks that use clean-label and mislabeled points to fail existing UDA methods and verify these attack methods in the experiments on two benchmark datasets.

However, there exists some limitations as follows.

1) In-depth analysis: despite the empirical discovery being interesting and promising, the current version seems preliminary and the in-depth analysis is lacking.

2) Insufficient theory: the proposed theoretical result is insufficient to understand the inherent limitation.

3) Writing issues: the writing in the manuscript is still problematic.

This paper is a boardline case according to the average rating. While the reviewers had some concerns on the significance, the authors did a particularly good job in their rebuttal. Thus, all of us have agreed to marginally accept this paper for publication! Please include the additional experimental results in the next version.